# Molecular models of multiple sclerosis severity identify heterogeneity of pathogenic mechanisms

Peter Kosa[1,3], Christopher Barbour[1,3], Mihael Varosanec[1], Alison Wichman[1], Mary Sandford[1], Mark Greenwood[2] & Bibiana Bielekova [1] ✉

While autopsy studies identify many abnormalities in the central nervous system (CNS) of subjects dying with neurological diseases, without their quantification in living subjects across the lifespan, pathogenic processes cannot be differentiated from epiphenomena. Using machine learning (ML), we searched for likely pathogenic mechanisms of multiple sclerosis (MS). We aggregated cerebrospinal fluid (CSF) biomarkers from 1305 proteins, measured blindly in the training dataset of untreated MS patients (N = 129), into models that predict past and future speed of disability accumulation across all MS phenotypes. Healthy volunteers (N = 24) data differentiated natural aging and sex effects from MS-related mechanisms. Resulting models, validated (Rho 0.40-0.51, p < 0.0001) in an independent longitudinal cohort (N = 98), uncovered intra-individual molecular heterogeneity. While candidate pathogenic processes must be validated in successful clinical trials, measuring them in living people will enable screening drugs for desired pharmacodynamic effects. This will facilitate drug development making, it hopefully more efficient and successful.

Effective management of chronic, polygenic diseases requires patient-specific polypharmacy regimens that target all pathogenic mechanisms underlying disease expression in the patient. This strategy is feasible, e.g., in cardiovascular diseases, where the contributing pathogenic mechanisms are easily measured. In contrast, it is currently impossible to measure diverse mechanisms that may mediate the destruction of the central nervous system (CNS). This limits new drug development and makes clinical management of patients suboptimal.

Advances in proteomics allow for accurate measurements of thousands of proteins in cerebrospinal fluid (CSF)[1,2]. These CSF proteins can be aggregated into molecular diagnostic test of multiple sclerosis (MS)[3] that outperforms magnetic resonance imaging (MRI)-based diagnosis of MS (i.e., independent cohort-validated area under receiver-operator characteristic curve (AUROC) 0.98 for the molecular diagnostic test[3] versus AUROC of -0.70 for the MRI-based tests[4]). In

recognition of the insufficient accuracy of MRI-based diagnosis, the 2017 revision of MS diagnostic criteria incorporates a possibility to evaluate CSF oligoclonal bands (OCB)[5]. This opens an opportunity to bring to clinical practice advanced laboratory tests that may pinpoint patient-specific pathophysiological drivers of CNS tissue damage, in addition to diagnosing a condition.

Pathologists identified multiple processes in MS CNS tissue autopsy but differentiating disease consequences from disease mechanisms is practically impossible when each patient can be analyzed only once, usually at the disease end. Intrathecally compartmentalized inflammation[6], associated with the tertiary lymphoid follicles, may be pathogenic based on correlations with rates of disability progression in a limited number of autopsy cases[7]. We recently validated relationship between intrathecal inflammation and MS severity in a prospectively acquired MS patients (N = 244); where CSF biomarkers of

[1]Neuroimmunological Diseases Section, National Institute of Allergy and Infectious Diseases, National Institutes of Health, Bethesda, MD, USA. [2]Department of Mathematical Sciences, Montana State University, Bozeman, MT, USA. [3]These authors contributed equally: Peter Kosa, Christopher Barbour. ✉e-mail: bibi.bielekova@nih.gov

intrathecal inflammation positively, but weakly (i.e., Rho = 0.18-0.24; $p$ = 0.044-0.002) correlate with the rates of disability progression[8].

Non-immune mechanisms such as mitochondrial dysfunction, hypoxia, oxidative stress, demyelination, toxic (A1) astroglial activation[9,10], and axonal transection might also be measured by CSF biomarkers. The most promising of these is neurofilament light chain (NFL)[11], detectable in healthy volunteers (HVs) but in greater quantities in neurodegenerative diseases. NFL correlates strongly with MS relapses or contrast-enhancing lesions (CELs) and has weak prognostic value for disability progression[12-16]. Additionally, NFL is an epiphenomenon reflective of ongoing axonal damage rather than its pathophysiological driver.

Thus, there remains a need for development of biomarkers reflective of diverse (ideally all) molecular intrathecal processes with potential pathogenic role in MS.

In this work we present CSF biomarker-based models of MS severity that provide insight into MS pathophysiology, identify molecular disease heterogeneity, and lead to an independent cohort-validated prognostic test(s).

## Results

The study design is depicted in Fig. 1. The collection of longitudinal clinical and cross-sectional brain MRI (Fig. 1a) volumetric outcomes is detailed in Methods.

Disability measured by clinical scales (Expanded Disability Status Scale [EDSS][17], Combinatorial Weight-adjusted Disability Score [CombiWISE][18]), or the amount of CNS tissue destruction reflected by brain parenchymal fraction (BPFr) increase with disease duration (DD) and patient's age (Fig. 1a). If these outcomes are changing with MS evolution, biological processes that correlate with these progression outcomes must also evolve intra-individually: i.e., be less expressed in patients with early MS (i.e., relapsing-remitting MS [RRMS]) and more prominent in patients with long disease duration and greater disability (i.e., progressive MS). These are processes expected to overlap with what pathologists identified in MS autopsies. While some of these evolving processes may contribute to CNS tissue destruction (i.e., might be pathogenic), others likely represent an epiphenomenon (i.e., inert) or even beneficial response of CNS to injury (i.e., protective).

To try to differentiate between potentially pathogenic, inert, or beneficial intrathecal processes, we can study which of them correlate with "MS severity", defined as the speed of disability progression. Ideally, we would study speed of disability accumulation in longitudinal cohorts. Practically longitudinal data are difficult to collect due to subject attrition. Diversity of treatments during longitudinal follow-up represents further impediment. Consequently, MS severity has been measured by cross-sectional outcomes that relate accumulated disability to either disease duration (in EDSS-based MS Severity Score [MSSS[19]]) or age (in EDSS-based Age-Related MS Severity Score [ARMSS[20]] and in CombiWISE-based MS Disease Severity Score [MS-DSS[21]]). As subclinical stage of MS may last years, relating disability to age is scientifically preferable, especially when epidemiological data suggests that MS starts in late childhood/early adulthood in most patients[22,23].

Age-based MS severity outcomes differentiate MS patients of identical age who accumulated more or less disability. As this comparison is done for all ages, biological processes that correlate with MS severity are unlikely to represent epiphenomena, because they occur equally in younger and older subjects. Instead, processes that correlate positively with MS severity are enriched in patients who accumulated disability faster; therefore, such processes might be pathogenic. Conversely, processes that correlate negatively with MS severity are candidate protective mechanisms, enriched in patients who accumulated disability slower.

This inference assumes that MS severity is relatively stable intra-individually in the absence of treatments. We can formally test intra-individual stability of MS severity by asking if past rates of MS progression reflected by cross-sectional MS severity outcomes predict future rates of MS progression (measured by longitudinal follow-up). Among 3 published MS severity scales, only MS-DSS was shown to predict future rates of disability progression in the independent validation cohort[21], likely because MS-DSS is based on CombiWISE[18], a continuous disability scale with much larger dynamic range than EDSS (i.e., ranging from 0-100). MS-DSS, in contrast to MSSS and ARMSS, also adjusts for multiple confounders, including the effect of applied disease modifying therapies (DMTs). We can further quantify intra-individual stability of MS-DSS by calculating intraclass correlation coefficient (ICC), which compares the fluctuation of longitudinal MS-DSS measurements for individual patients with the variance of MS-DSS measured between MS patients. The ICC close to 1 indicates complete interchangeability of intra-individual measurements (i.e., patient-specific MS-DSS does not fluctuate), whereas value close to 0 indicates high fluctuation of MS-DSS values in repeated measurements. The ICC for MS-DSS is 0.90 (Fig S1).

Validating intra-individual stability of MS-DSS allows us to link any MS-DSS measurement to CSF sample collected from the same patient. We selected the MS-DSS calculated at the first untreated clinic visit (concomitantly with CSF collection; Fig. 1a) as the primary outcome against which we modeled CSF biomarkers, as this allowed us to test the hypothesis that CSF biomarker-based model of MS-DSS will predict future rates of disability progression measured from subsequent clinic visits. As sensitivity analyses for the robustness of the gained biological insight, we used MS-DSS collected at the last clinical follow-up, as secondary outcome. In 2017 (which falls between first and last clinic visit for most subjects) we developed the NeurEx™ App[24]. NeurEx™ eliminates scoring differences among clinicians by algorithmically computing disability scales from clinician-documented examination. We hypothesized that by eliminating this source of noise, MS-DSS computed from NeurEx™ scores will be more accurate, leading to CSF-biomarker model that reflects overlapping biology with the model of primary outcome, but achieves higher effect size. We also hypothesized that MS-DSS models will predict EDSS-based MS severity outcomes, especially ARMSS, which shares the age denominator with MS-DSS.

Finally, as an exploratory outcome we wanted to assess biology associated with rates of CNS tissue destruction, using cross-sectional outcome analogous to disability-based MS severity outcomes. Brain volume deficit (BVD) severity outcome, calculated as residuals from the linear regression model of 1-BPFr against age (Fig. 1a) was calculated from a single brain MRI performed within 3 months of CSF collection. Patients with higher BVD severity have lost more brain tissue than their equally aged peers.

### Adjusting SOMAmers based on physiological age and sex associations (Fig. 1b)

Some of the processes that pathologists identified in MS brain autopsies overlap with processes associated with natural aging: e.g., mitochondrial dysfunction, oxidative stress or activation of innate immunity[25-28]. Without access to HV data it would be impossible to determine if processes that correlate with age in MS cohort represent physiological aging, MS-related mechanisms, or both. This is important, as MS DMTs are unlikely to inhibit physiological aging.

Therefore, we sought to differentiate the natural aging (and physiological sex differences) from MS-specific processes using HV CSF data (Fig. 1b). As our HV cohort was small ($N$ = 24; Table 1), we applied 2-tier analyses (Fig. 1b) to conserve $p$-values by including prior knowledge. Hypothesizing that aging exerts same effect on proteins measurable in serum and CSF, in the first analysis we prioritized biomarkers that already showed strong relationship with age in a published cohort of 3301 HV from the INTERVAL study analyzing serum by identical DNA-aptamer-based SomaScan® technology[1]. Specifically, we assessed: A. Concordant directionality in the relationships (p < 0.05) between INTERVAL HV and our HV CSF cohort; and B. Statistically

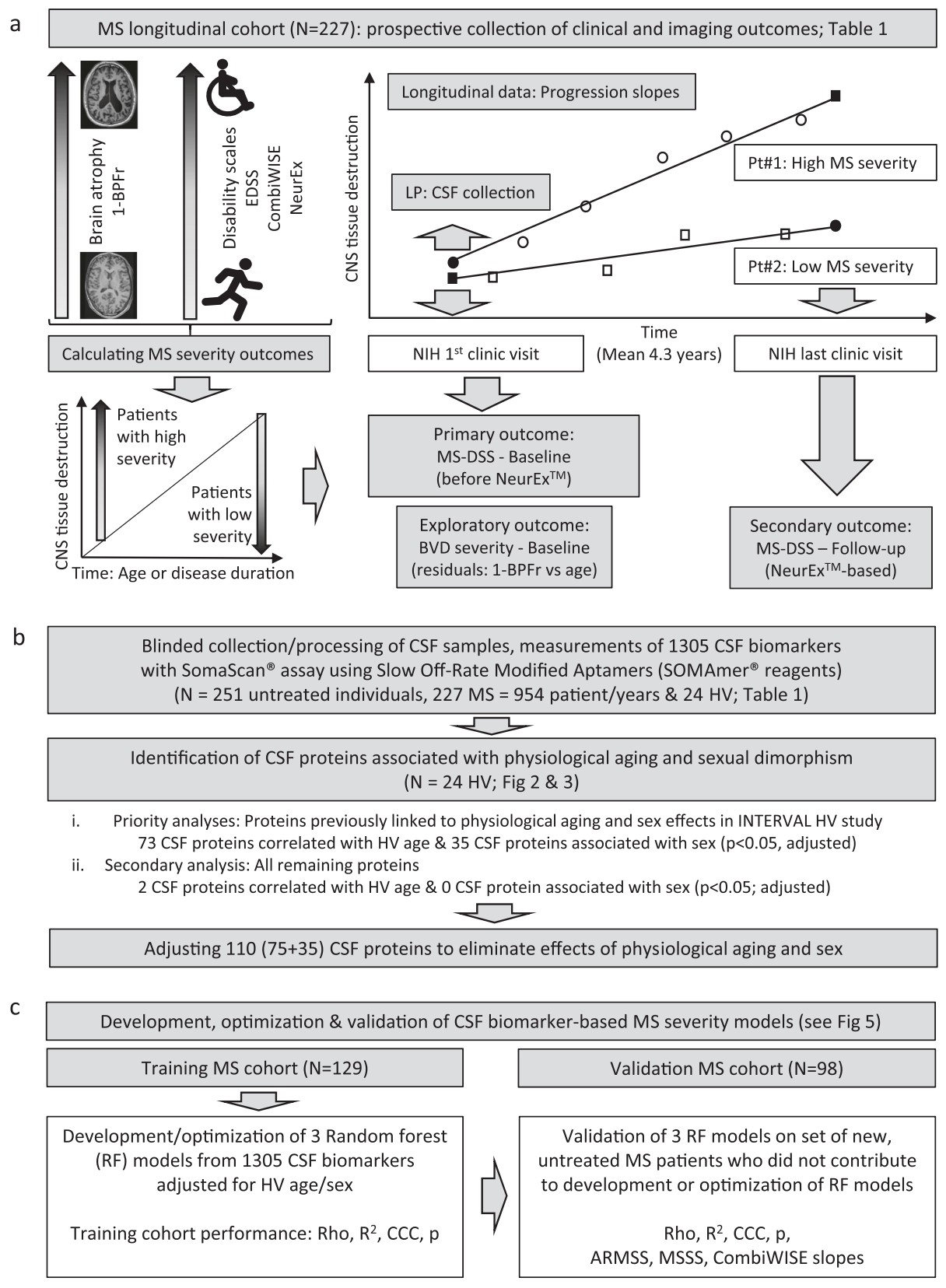

**Fig. 1 | Experimental design. a** Prospective collection of longitudinal clinical (Expanded Disability Status Scale [EDSS], Combinatorial Age-adjusted Disability Score [CombiWISE]), and cross-sectional imaging outcomes (brain parenchymal fraction (BPFr]) paired with lumbar puncture (LP) at the first clinic visit. **b** 1305 biomarkers measured in blinded fashion in cerebrospinal fluid (CSF) samples of multiple sclerosis (MS) patients and healthy volunteers (HV) were mathematically adjusted to eliminate the effects of aging and sex. **c** Random forest (RF) algorithm was applied to training cohort ($N = 129$) data, resulting in three models of MS severity. Models' performance was assessed by Spearman Rho, $R^2$, Concordance Correlation Coefficient (CCC), and $p$-value ($p$) of the Spearman correlation between observed and model-predicted values. The validity of the three models was then evaluated in an independent cohort ($N = 98$) by measuring the above-mentioned characteristics of the observed vs predicted outcomes. MS-DSS Multiple Sclerosis Disease Severity Scale, BVD brain volume deficit.

**Table 1 | Demographic data for the training, validation, and HV cohorts**

|  | Controls | Training cohort | | | Validation cohort | | | *p*-value |
|---|---|---|---|---|---|---|---|---|
|  | HV | RRMS | SPMS | PPMS | RRMS | SPMS | PPMS |  |
| *N* (female/male) | 24 (11/13) | 37 (19/18) | 31 (21/10) | 61 (29/32) | 33 (20/13) | 24 (15/9) | 41 (19/22) | 0.915 |
| Average Age (SD) | 40.9 (11.4) | 40.9 (11.1) | 52.3 (9.0) | 54.8 (7.9) | 39.5 (9.5) | 51.9 (12.2) | 54.7 (11.3) | 0.583 |
| Average DD (SD) | NA | 4.8 (6.7) | 22.4 (9.9) | 11.7 (8.2) | 6.0 (7.7) | 19.6 (10.7) | 12.8 (8.5) | 0.989 |
| Average EDSS (SD) | NA | 1.8 (1.2) | 5.9 (1.2) | 5.3 (1.6) | 2.2 (1.6) | 5.5 (1.5) | 5.2 (1.6) | 0.610 |
| Average MS-DSS (SD) | NA | 1.3 (0.5) | 2.1 (1.1) | 2.0 (0.8) | 1.4 (0.7) | 2.3 (1.3) | 1.9 (1.0) | 0.511 |

*p*-value column tests for differences in demographic parameters between the two cohorts (excluding controls), using a chi-squared test for sex and a Wilcoxon rank test for quantitative variables. All statistical tests were two-sided. See also Methods section.

*HV* healthy volunteer, *RRMS* relapsing-remitting multiple sclerosis, *SPMS* secondary progressive multiple sclerosis, *PPMS* primary progressive multiple sclerosis, *EDSS* expanded disability status scale, *DD* disease duration, *MS-DSS* Multiple Sclerosis Disability Severity Score, *SD* standard deviation. See also Methods section.

significant relationship with age and/or sex in our MS cohort (demographic data available in Table 1).

Using this approach, 73 age-associated biomarkers had adjusted *p* < 0.05 (Fig. 2). Considering that some CSF proteins may not be measurable in the serum, we also assessed correlation with age and sex for remaining biomarkers not prioritized above. This identified two additional proteins (PGF and SLPI) in our HV CSF cohort with evidence of age associations after Bonferroni adjustments. Out of these 75 HV age-associated biomarkers, 22 (29.3%) showed discordant changes between the HV and the MS cohort (i.e., increasing with age in CSF of MS patients and decreasing with age in HV) (Fig. 2a).

On the example of GDF15, the validated biomarker of mitochondrial dysfunction[29–32], Fig. 2b showcases the difference between subtracting only HV-aging variance from the CSF protein levels and regressing age as covariate based on MS cohort only, which is the usual way to adjust for confounding effects. The Fig. 2b left panels demonstrate that CSF GDF15 correlates with age both in HV (top panel, blue color) and MS cohorts (bottom panel, black color), even though distribution of MS values suggests elevation of GDF15 beyond physiological aging with MS progression. This is validated in right panels, where regressing out only physiological aging demonstrates residual positive correlation of HV-Age-adjusted GDF15 CSF levels with MS age ($R^2 = 0.1$, $p = 7.4 \times 10^{-6}$). Thus, we conclude that while mitochondrial dysfunction is associated with physiological aging, there is additional, MS-related mitochondrial dysfunction that increases with MS progression. This conclusion is consistent with published pathological observations in MS[33]. Regressing out age in MS cohort as covariate would fail to identify mitochondrial dysfunction beyond natural aging associated with MS. Conversely, ignoring age altogether would overestimate the amount of mitochondrial dysfunction linked to MS.

To verify that identified proteins are indeed age-related based on current knowledge, we used the Search Tool for the Retrieval of Interacting Genes/Proteins (STRING) annotations. Reassuringly, this analysis (Fig. 2 and Supplementary Data 1) identified Kyoto Encyclopedia of Genes and Genomes (KEGG) and Reactome pathways previously associated with physiological aging, such as proteoglycans/chondroitin sulfate and extracellular matrix reorganization, signaling pathways p53, PI3K-AKT, MAPK, HIF-1 and WNT, apoptosis, and Alzheimer's disease. While most of the age-concordant proteins were proteins secreted to extracellular space and were part of the extracellular matrix, age-discordant CSF proteins (i.e., decreased in HV but increased in MS) belonged to two categories: Secreted proteins linked to immune system; and the cell surface/membrane-anchored proteins found in axons and the neuronal cell body (Fig. 2 and Supplementary Data 2). This suggests re-expression of these neuronal receptors and pathways in MS or their release by MS-associated neuronal injury. The pathways enriched for Age-discordant CNS proteins are metabolism, axon guidance, netrin-1, NOTCH, hedgehog, and thyroid hormone signaling, all linked to neurogenesis or myelination.

Using the same strategy, 35 biomarkers were linked to physiological sex differences in CSF, with all but one (SERPINA10) showing concordant differences between MS patients and HV (Fig. 3). STRING analysis confirmed validity of our approach: the seven proteins elevated in females are related to ovulation, ovarian steroidogenesis, and prolactin signaling (Fig. 3 and Supplementary Data 3). Male-elevated proteins are linked to immunity (innate immunity, chemokines), fluid shear stress, and atherosclerosis (Fig. 3 and Supplementary Data 4), consistent with the reported effects of Y-chromosome genes on inflammation and atherosclerosis[34].

For all downstream analyses we used HV age- and sex-adjusted values for 110 proteins with significant physiological confounding effects (Fig. 1b).

## MS is not associated with accelerated aging

Among the proposed hypotheses of MS progression is the idea that MS patients suffer from accelerated aging[35]. Thus, we tested the hypothesis that the CSF proteomic signature of physiological aging estimates higher than biological age for MS patients. To this end, we used a regularized multiple linear regression (elastic net) to develop a CSF biomarker-based model of chronological age in HV cohort (Fig. 4a). When we used this model to predict chronological age in MS patients (Fig. 4b), we did not observe evidence for accelerated aging. Instead, the model slightly overestimated age in RRMS (without reaching statistical significance). Surprisingly, the model underestimated physiological age in both progressive MS subtypes (Fig. 4c). Thus, we conclude that molecular mechanisms different from physiological aging are responsible for CNS tissue loss in MS, at least as reflected by CSF proteins measured in this study.

## Identifying molecular pathways associated with MS severity

To gain biological insight about processes that correlate with MS severity, we used two Functional Enrichment Analyses (FEA) (Fig. 5). FEA uses associations of all measured CSF proteins with MS severity outcomes (MS-DSS at baseline, MS-DSS at follow-up, and BVD severity): either captured by correlation coefficients (for STRING ordered analysis[36]) or by false discovery rate (FDR)-adjusted *p*-values (for g:Profiler ordered analysis[37]). To increase FEA stringency, we focused on those processes/pathways that achieved FDR-corrected statistical significance in both g:Profiler and STRING FEA. While all gene ontology (GO) terms and REACTOME pathways (and their contributing CSF biomarkers) that fulfilled these pipeline criteria are provided in Supplementary Data 5, based on the overlap of the contributing CSF biomarkers, we merged GO/REACTOME terms into five distinct biological categories (Fig. 5; left panels).

As we hypothesized, we observed strong overlap in biological processes that correlated with MS-DSS measured at first and last clinic visit. Surprisingly, somewhat different biological processes were associated with imaging BVD severity outcome: The coagulation cascade was only associated with BVD severity outcome and Complement

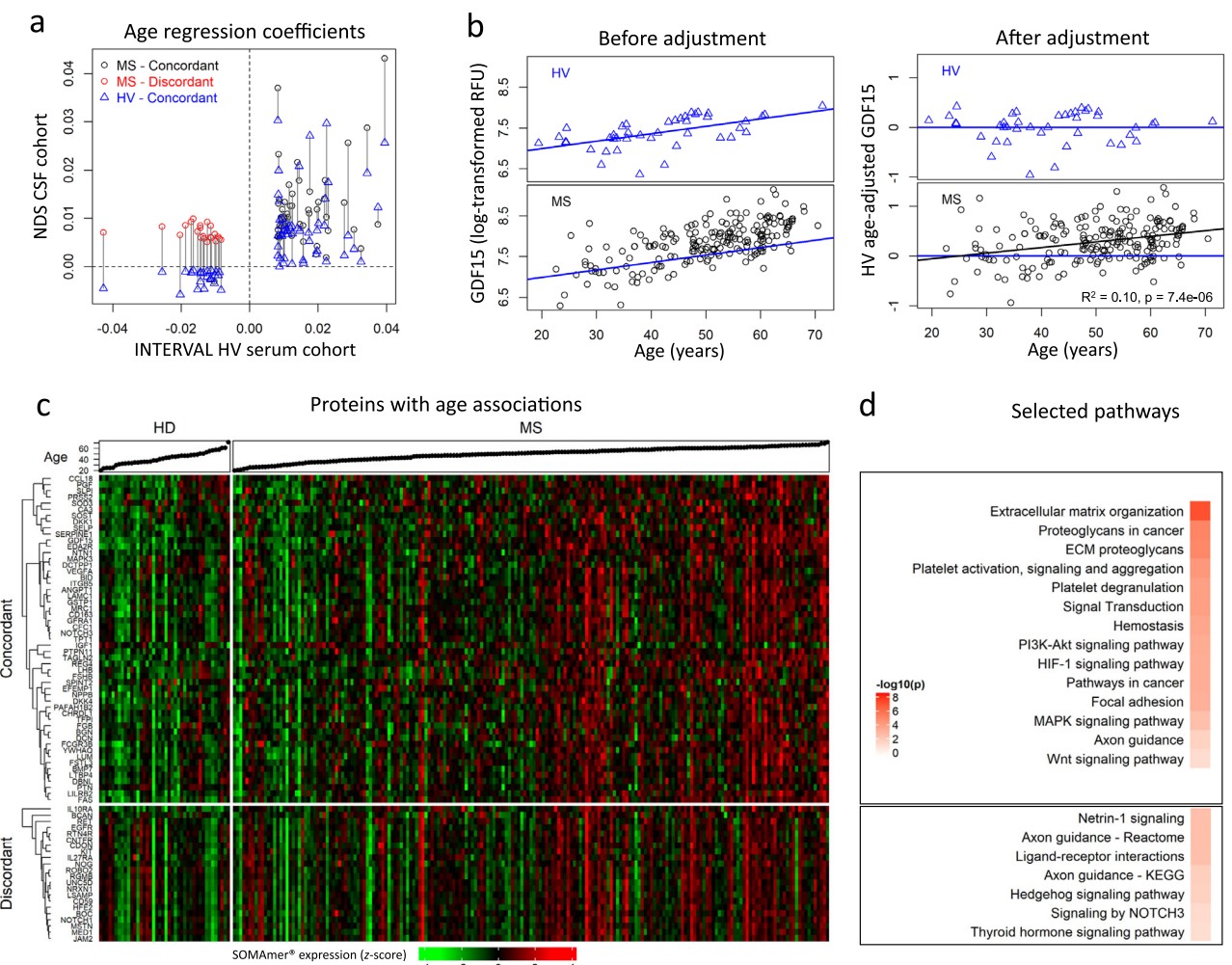

**Fig. 2 | Adjusting SOMAmers based on physiological age associations.**
**a** Regression coefficients for the 75 SOMAmers with age associations verified in healthy volunteers (HV) cerebrospinal fluid (CSF). Blue triangles compare effect sizes (regression coefficients) of physiological age on protein concentrations in serum (external HV cohort from INTERVAL study; x-axis) versus CSF (internal HV cohort; y-axis). Circles correspond to multiple sclerosis (MS) CSF coefficients with concordant (black) or discordant (red) associations with age compared to HV cohorts. Vertical lines connect the CSF coefficients for MS and HV cohorts for the same biomarker. **b** Example of adjusting measured CSF concentrations of a single protein (growth differentiation factor 15; GDF15) by subtracting effect of healthy aging. GDF15 log-transformed relative fluorescent unit (RFU) values (y-axis) versus age (x-axis) are displayed for HV (top) and MS (bottom) cohorts, before (left) and after (right) adjustment. The HV simple linear regression line (blue) used for the adjustment is superimposed on each panel. The coefficient of determination $[R^2]$ and the corresponding p-value were extracted from the linear model (represented by the black line) of HV age-adjusted GDF15 values versus age in MS patients. **c** Heatmap displaying the standardized expression (log-scaled z-scores) for the 75 selected SOMAmers (rows, for ordered list of proteins, see Supplementary Data 15), separated based on HV/MS concordance or discordance, for all patient samples (columns). Corresponding ages for each participant are displayed in ascending order at the top of the heatmap. **d** Selected pathways identified using functional enrichment STRING analysis along with Benjamini–Hochberg-adjusted −log10(p-values) describing how significant the functional enrichment is for age concordant and discordant proteins, respectively. See also Supplementary Data 1 and Supplementary Data 2. All statistical tests were two-sided. Source data are provided as a Source Data file.

cascade, while significantly associated with all three outcomes showed lower p-values and more than twice contributing GO/REACTOME complement-related terms with BVD severity as compared to MS-DSS. In contrast, NOTCH signaling (specifically, NOTCH1 and NOTCH3, JAG1, JAG2, DLL1, DLL4) was significantly associated only with MS-DSS outcomes. The "Neuron recognition" category, enriched for proteins involved in Ephrin signaling, neuronal recognition, junctional molecules, and axon guidance proteins were associated with all three MS severity outcomes, with stronger MS-DSS association based on lower p-values and higher number of significant terms.

To provide directionality of these biological categories with MS severity outcomes, we aggregated either positively or negatively correlated CSF-biomarkers (FDR-adjusted p < 0.05) with MS severity outcomes and ran g:Profiler enrichment analysis using operator-defined background of the 1305 proteins included in the SOMASCan (Fig. 5;

right panels). This analysis demonstrated positive associations of coagulation and complement cascades and negative associations for NOTCH signaling and neuron recognition categories with MS severity. As the proteins from Innate immunity/cytotoxicity category had both positive and negative correlations with MS severity outcomes, this category did not exert statistically significant positive or negative associations with MS severity.

Spearman correlation coefficients and FDR-adjusted p-values[38] for all individual CSF biomarkers are in the Supplementary Data 6. We observed large differences in the number of CSF proteins that were significantly (FDR-adjusted) correlated with different MS severity outcomes: 26 for MS-DSS measured at baseline, 76 for MS-DSS at follow-up and 55 for BVD severity. Only two SOMAmers correlated with ARMSS at baseline and one at follow-up visits and no biomarkers correlated with MSSS. Each of these CSF proteins showed only small

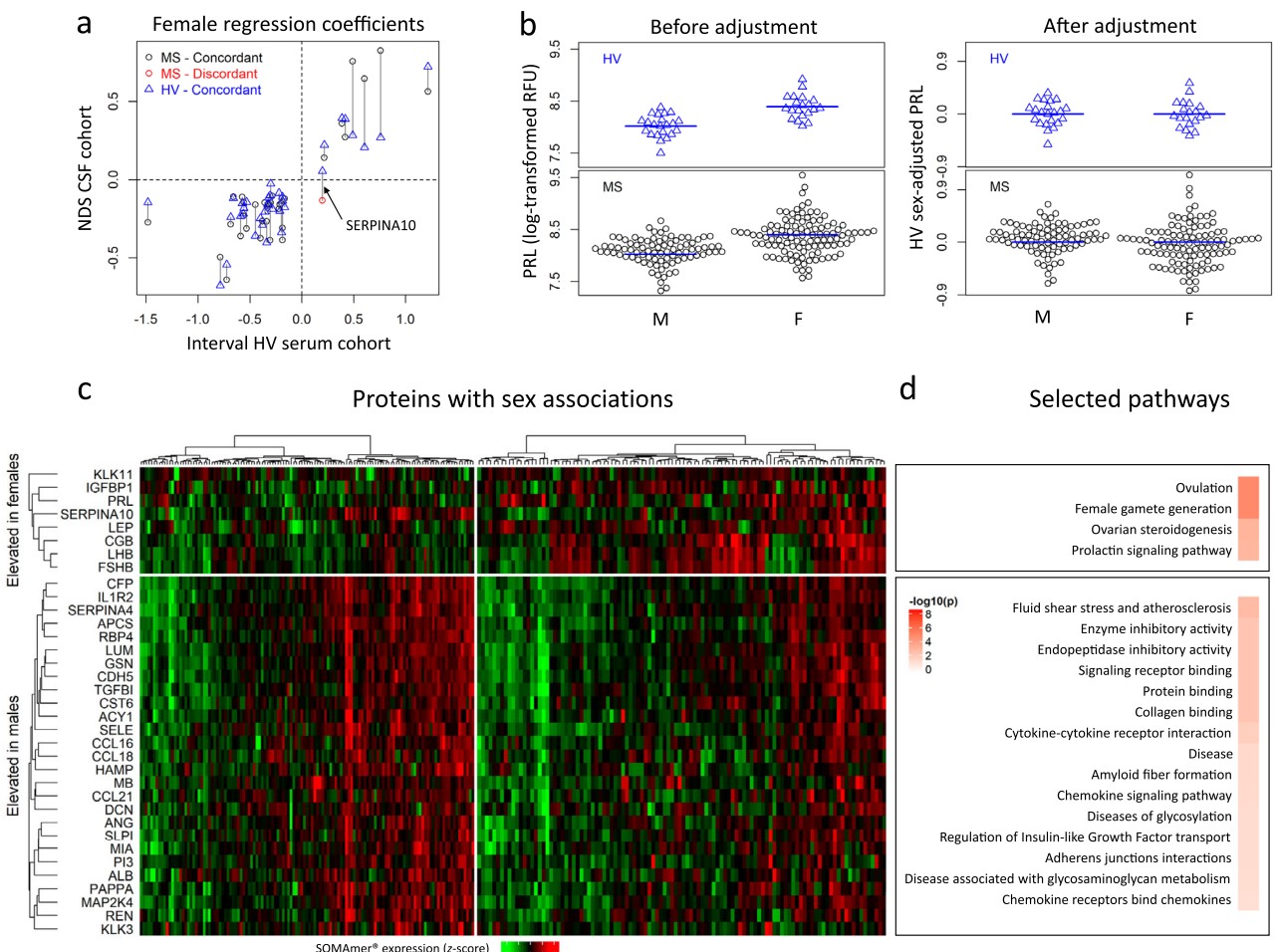

**Fig. 3 | Adjusting SOMAmers to subtract effects of physiological sex associations. a** Regression coefficients for the 35 SOMAmers with sex associations from cerebrospinal fluid (CSF) and serum. Blue triangles compare effect sizes (regression coefficients) of protein association with sex measured in healthy volunteer (HV) serum in the published (INTERVAL) study (x-axis) with internal HV CSF cohort (y-axis). Circles correspond to multiple sclerosis (MS) CSF coefficients with concordant (black) and discordant (red) associations with sex compared to HV. Vertical lines connect the CSF coefficients for our MS and HV cohorts for the same biomarker. SERPINA10, identified by a black arrow, showed discordant association with sex in MS versus HV. **b** Example of adjusting CSF protein concentration to subtract effects of physiological sex differences on prolactin (PRL).

CSF PRL log-transformed relative fluorescent unit (RFU) values (y-axis) versus sex (x-axis) are displayed for both HVs (top) and MS (bottom) cohorts, before (left) and after (right) adjustment, showing no residual difference between MS and HV. **c** Heatmap displaying the standardized expression (log-scaled z-scores) for the 35 sex-associated biomarkers (rows), separated based on elevation in females/males, for all patient samples, separated by males and females (columns). **d** Selected pathways identified using functional enrichment STRING analysis along with Benjamini−Hochberg-adjusted −log10(p-values) describing how significant the enrichment is for female-elevated and male-elevated proteins, respectively. See also Supplementary Data 3 and Supplementary Data 4. All statistical tests were two-sided. Source data are provided as a Source Data file.

effect size when correlating with MS severity outcomes (i.e., up to Spearman Rho = 0.382 for MMP7).

**Development and validation of CSF biomarker-based MS severity models (Fig. 1c)**

Observing that only few CSF proteins correlated significantly with MS severity outcomes and all exerted small effect sizes, we asked whether we can use machine learning (ML; i.e., random forest[39] with a variable selection pipeline[40]) to aggregate CSF biomarkers into models that predict MS severity in the independent validation cohort with effect sizes higher than any single CSF biomarker (Figs. 1c and 6a).

For the primary outcome (MS-DSS at baseline; Fig. 6b), the model selected 57 SOMAmer ratios (75 unique biomarkers) and explained 62% of variance in the training cohort (Fig. 6b, left panel, Rho = 0.767, $R^2 = 0.618$, CCC = 0.662 [CCC = Concordance Correlation Coefficient-reflects 1:1 fit between measured and CSF-predicted outcomes, with perfect fit = 1]; $p < 2.2 \times 10^{-16}$). 21 ratios (34 unique SOMAmers) selected by MS-DSS model based on follow-up clinical data (secondary outcome)

had the strongest training cohort effect size (MS-DSS Follow-up; training cohort results [Fig. 6c, right panel]: Rho = 0.781, $R^2 = 0.634$, CCC = 0.719; $p < 2.2 \times 10^{-16}$). The BVD severity model (exploratory outcome), consisting of 21 ratios (35 unique biomarkers) explained 60% of variance (Fig. 6b, middle panel, Rho = 0.778, $R^2 = 0.597$, CCC = 0.675; $p < 2.2 \times 10^{-16}$). Collectively 3 MS severity models used 99 SOMAmer ratios; 97 unique and two SOMAmer ratios shared between the models predicting MS-DSS at baseline and at follow-up.

Considering the small number of CSF biomarkers that constitute each of these models (i.e., representing 0.1−0.3% of human proteome), the effect sizes observed in the training cohort were almost certainly too optimistic. ML-based algorithms invariably overfit the data and the amount of overfit cannot be determined unless the models are applied to new observations (independent validation cohort) not used in the model development (Fig. 6c).

When applied to validation cohort, all three models validated with very low p-values. Expectedly, the effect sizes diminished considerably; The CSF-based MS-DSS at baseline model captured 17% of variance of

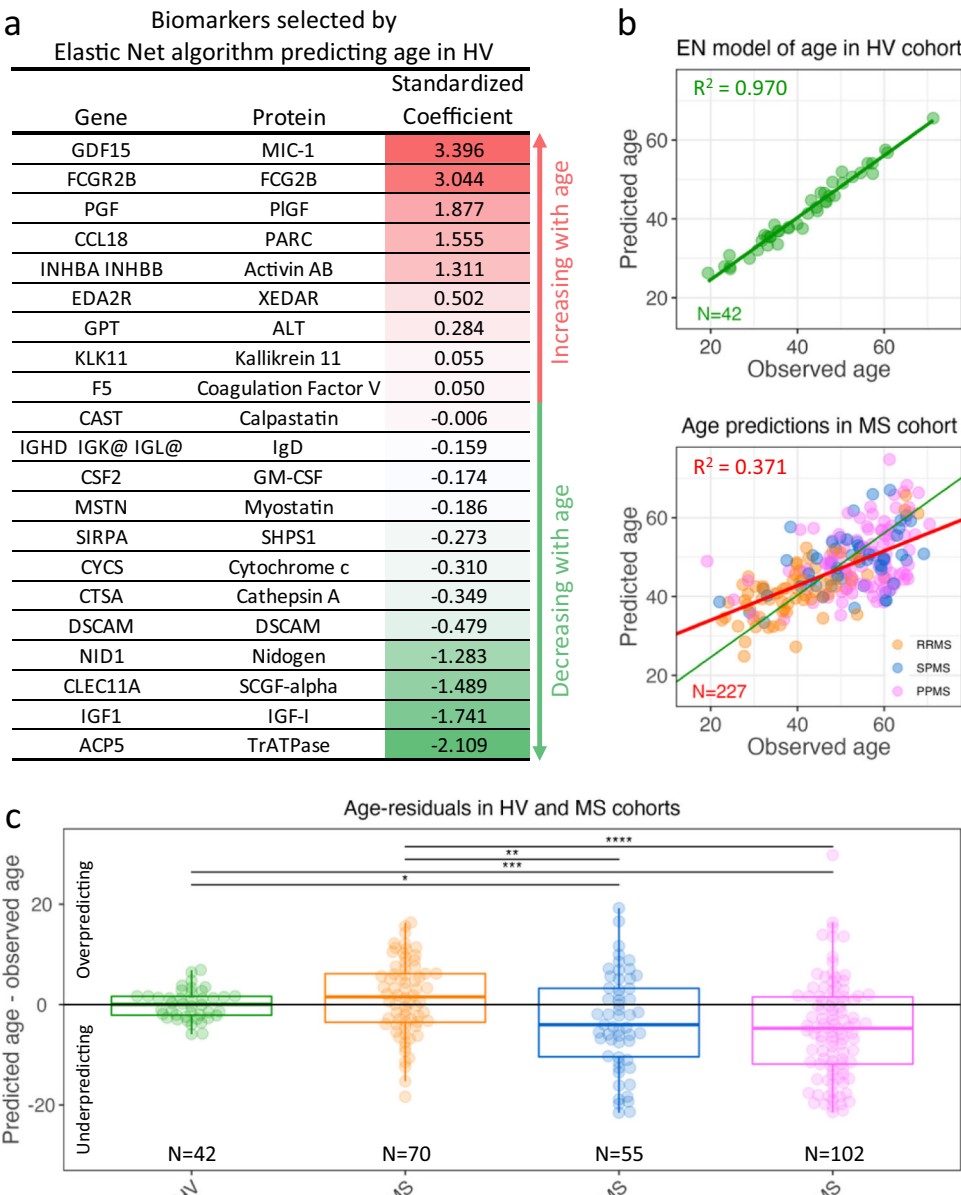

**Fig. 4 | MS is not associated with accelerated aging. a** Standardized regression coefficients from the elastic net (EN) model predicting age in the healthy volunteers (HV). Red-shading corresponds to biomarkers that increase with age, and green-shading corresponds to biomarkers decreasing with age. **b** Observed vs model-predicted age in the HV cohort (top) and multiple sclerosis (MS) cohort (bottom). The linear regression line (red) of observed vs predicted MS samples is superimposed on the green regression line of the HV cohort. The coefficient of determination ($R^2$) of the red line shows that cerebrospinal fluid (CSF) biomarkers explain almost 40% of variance associated with age of MS patients. **c** Difference between CSF model-predicted ages and observed ages (y-axis) in HV and MS subtypes (x-axis). The black bars mark significant differences based on pairwise

comparisons of the diagnostic groups using two-sided Wilcoxon test and false-discovery rate (FDR) adjustment for multiple comparisons ($p < 0.0001$ ****, $p < 0.001$ ***, $p < 0.01$ **, $p < 0.05$ *). Exact FDR-adjusted p-values for individual comparisons: HV-SPMS: $p = 0.049$, HV-PPMS: $p = 0.00058$, RRMS-SPMS: $p = 0.0072$, RRMS-PPMS: $p = 2.3 \times 10^{-5}$. The lower and upper hinges of the boxplots correspond to the first and third quartiles (the 25th and 75th percentiles). The upper whisker extends from the hinge to the largest value no further than 1.5 * interquartile range (IQR) from the hinge. The lower whisker extends from the hinge to the smallest value at most 1.5 * IQR of the hinge. Source data are provided as a Source Data file.

measured MS-DSS (Fig. 6c, left panel, Rho = 0.395, $R^2 = 0.166$, CCC = 0.306; $p = 6.5 \times 10^{-5}$), BVD severity model captured 22% of variance of measured values (Fig. 6c, middle panel, Rho = 0.470, $R^2 = 0.219$, CCC = 0.400; $p = 1.1 \times 10^{-5}$) and MS-DSS at follow-up captured 26% of variance (Fig. 6c, right panel, Rho = 0.505, $R^2 = 0.264$, CCC = 0.430; $p = 2.4 \times 10^{-7}$). This hierarchy of model validation (i.e., MS-DSS at baseline < BVD severity < MS-DSS at follow-up) was identical to the hierarchy with which outcomes correlated with individual CSF proteins.

Supplementary Data 7 contains annotated workbook that includes variable importance metrics[41] for all three models.

## CSF biomarker-based model predicts future rates of disability accumulation, as well as EDSS-based MS severity outcomes

The validated CSF biomarker-based models explain between 17 and 26% of variance measured by MS severity outcomes. How should we interpret this performance and how does it compare to published biomarkers/models of MS severity?

First, it is important to dissect plausible relationship between modeled outcomes (i.e., MS severity scales) and modeling predictors (i.e., CSF proteins). The biological substrate of neurological disability is loss of neuronal functions, molecularly reflected by transient loss of

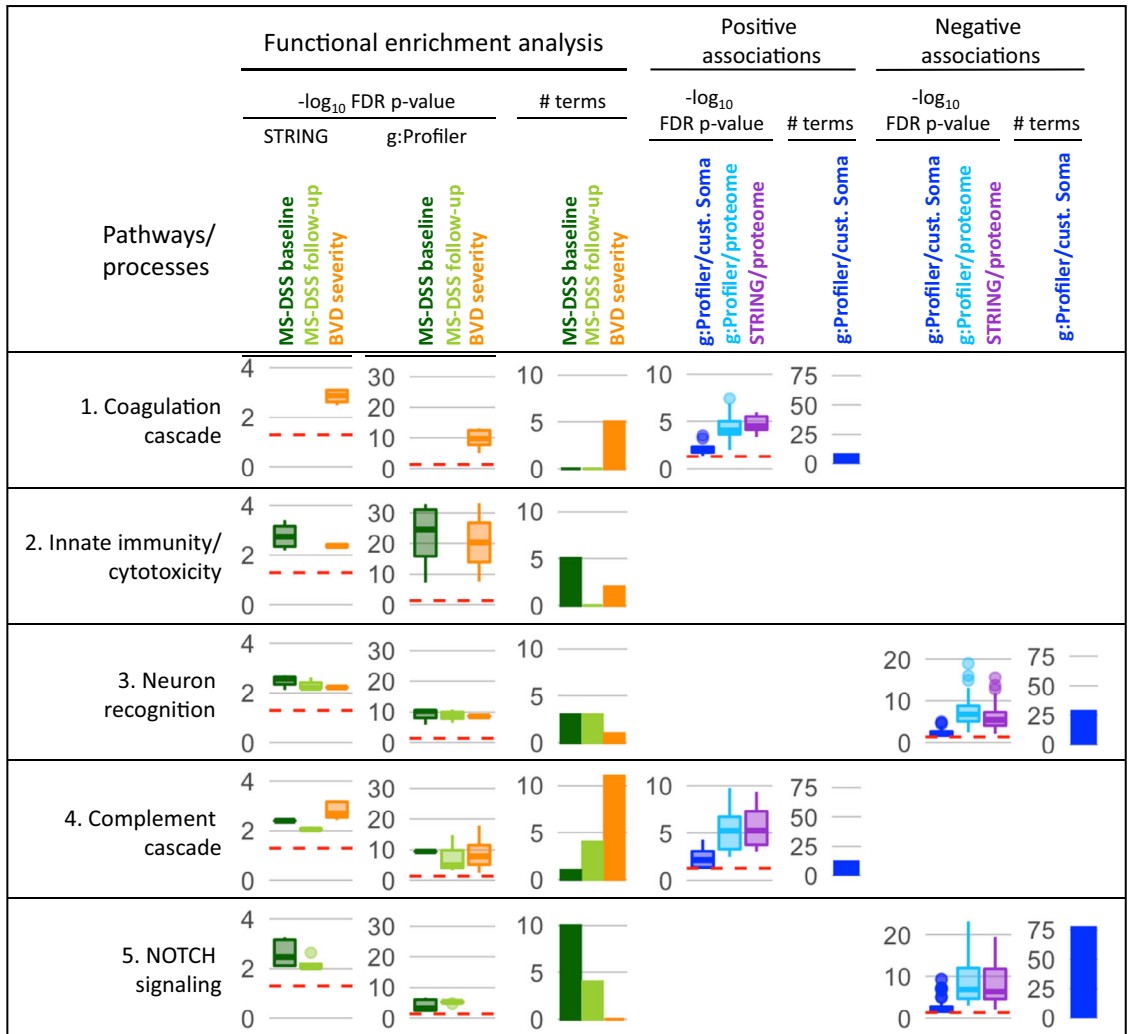

**Fig. 5 | Molecular pathways associated with MS severity.** One-thousand three-hundred five healthy volunteer (HV) age- and sex-adjusted SOMAmers were correlated with the three multiple sclerosis (MS) severity outcomes—multiple sclerosis disease severity score (MS-DSS) at baseline (dark green), MS-DSS at follow-up (light green), and brain volume deficit (BVD) severity (orange). Spearman correlation coefficients were used for the Functional enrichment analysis (FEA) in the STRING database. Enriched pathways and processes with false-discovery rate (FDR)-adjusted $p$-value < 0.05 were grouped into five main categories, and the boxplots for the $p$-values of individual processed are displayed. The validity of the findings was tested in g:Profiler database, where the same list of 1305 genes ordered by the increasing $p$-value was inputted for the FEA using the g:GOSt tool. The boxplots of FDR-adjusted $p$-values are shown. # term counts the number of processes identified for each category and outcome. Biomarkers significantly (FDR-adjusted $p$-value < 0.05) correlating with either of the three outcomes were submitted to g:Profiler using the custom set of 1305 SOMAmers (dark blue) or the whole proteome (light blue) as analysis background. The same set of SOMAmers was also analyzed by STRING using whole proteome background (violet). Boxplots of $p$-values for significantly enriched processes are displayed, as well as the number of significantly enriched processes that the g:Profiler identified linked to MS severity outcomes using 1305 SomaScan proteins as a background. FDR-adjusted $p$-values are displayed on a $-\log_{10}$ scale, the red dashed line depicts the FDR-adjusted $p$-value of 0.05. The lower and upper hinges of the boxplots correspond to the first and third quartiles (the 25th and 75th percentiles). The upper whisker extends from the hinge to the largest value no further than 1.5 * interquartile range (IQR) from the hinge. The lower whisker extends from the hinge to the smallest value at most 1.5 * IQR of the hinge. Source data are provided as a Source Data file.

electrical conductivity due to inflammation and associated blood–brain barrier (BBB) opening, demyelination, lack of glial support, pathological synaptic pruning and eventually death of neurons. These heterogeneous processes might be captured by CSF proteome (see below), but they can't be differentiated by clinical (or imaging) severity scales.

In other words, as our measurements do not capture complexity of underlying process, it is impossible to measure MS severity using clinical or imaging outcomes with the precision comparable to measuring physical phenomena, such as distance between two points in physical space. Different tools that measure physical distance will capture close to 100% variance, irrespective of measuring units they use. In contrast, the correlation matrix (Fig. 7a and Supplementary Data 8) shows only modest correlations between MS severity outcomes measured at first clinic visit, explaining minimum of 0, maximum of 55 and an average of

16% of variance in the independent validation cohort. If the clinical measurements of MS severity explain up to 55% of variance, it is impossible for CSF biomarkers to explain more.

If the correlation between MS severity outcomes is limited, how can we judge which outcome is most relevant? We can assess clinical value of MS severity outcomes by measuring effect sizes with which they predict future rates of disability progression. Only MS-DSS (but not MSSS or ARMSS) predicted future rates of disability progression in the independent validation cohort, measured prospectively by CombiWISE and adjusted for the effect of treatments as described[21] (Fig. 7a). Reassuringly, CSF biomarker-based model of MS-DSS (i.e., primary outcome) also predicted future rates of disability progression in the independent validation cohort with comparable (i.e., Rho = 0.26, $p$ = 0.0175 FDR-adjusted) effect size as clinical (MS-DSS) outcome. Even the

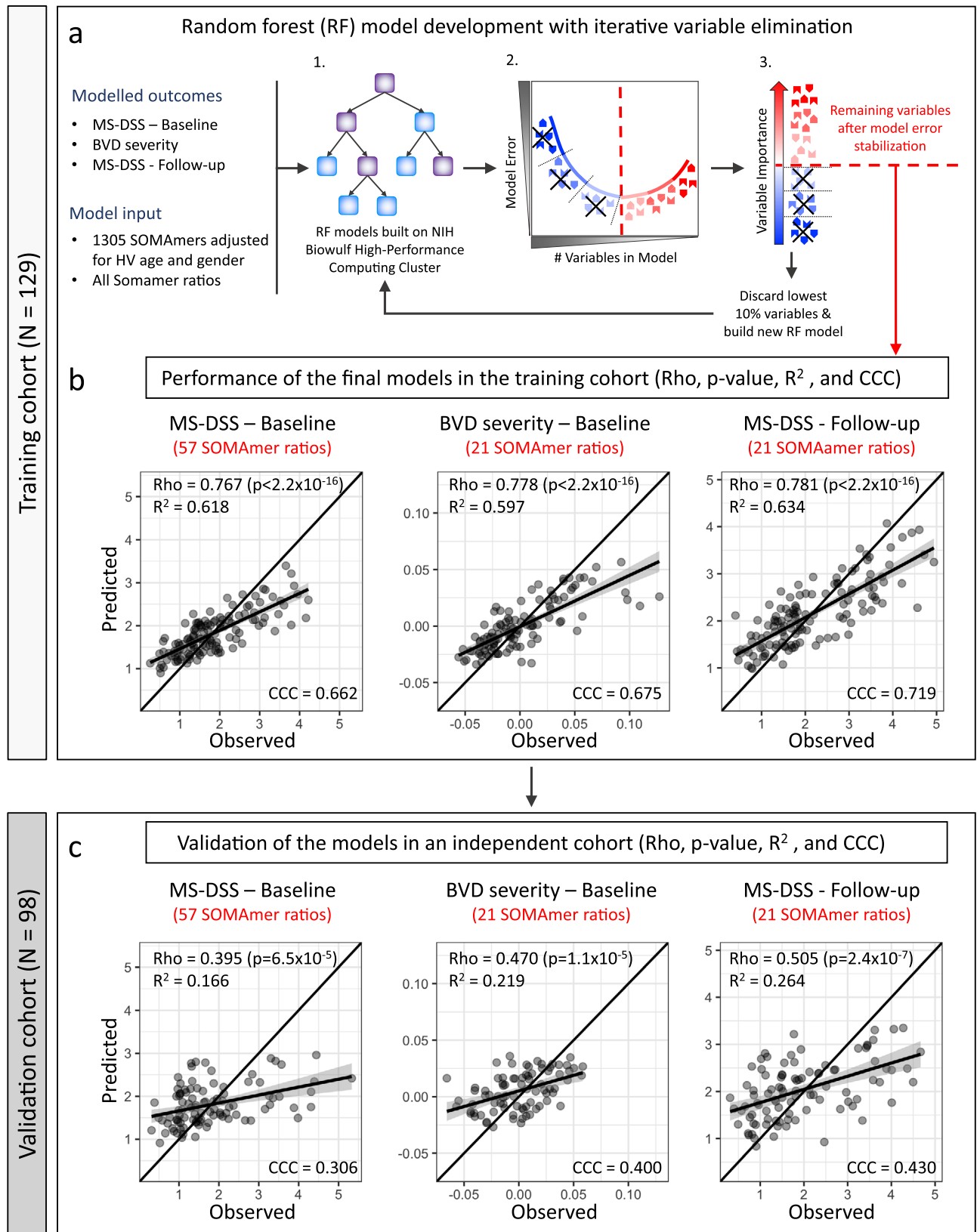

CSF biomarker-based model of BVD severity predicted future rates of MS disability progression with higher effect size than EDSS-based MS severity outcomes (Rho = 0.21), but the *p*-value was no longer statistically significant after adjusting for multiple comparisons (*p* = 0.06). Although CSF biomarker-based model of MS-DSS collected at last clinic visit also correlated with MS the progression slopes (Supplementary Data 8), this comparison contains a circular argument, in that MS-DSS measured at last clinic visit already comprises the disability progression that occurred during follow-up. We conclude that CSF biomarker-based models out-performed EDSS-based MS severity outcomes (MSSS and ARMSS) and matched MS-DSS in predicting future slopes of disability progression in the independent validation cohort.

**Fig. 6 | Development and validation of CSF-based MS severity models. a** All models were developed and optimized in the training cohort (N = 129). Three modeling outcomes were used: Multiple Sclerosis Disease Severity Score (MS-DSS) at baseline, brain volume deficit (BVD) severity at baseline, and MS-DSS at most recent follow-up. Healthy volunteer (HV) age- and sex-adjusted SOMAmers and all possible SOMAmer ratios were used as variables for the modeling. Random forest models were generated using a high-performance computing cluster (1), A statistical learning pipeline optimized models by decreasing the number of predictors to minimize overfit: At each step, we constructed 10 random forest models and recorded the training out-of-bag (OOB) model error (2), We also averaged variable importance measures from these 10 random forest models based on node impurity (3). The 10%

least contributing variables were excluded, and the process repeated till the OOB error had minimized (red dashed line). The remaining predictors constituted the final/optimized model. **b** Performance of the final models was evaluated by Spearman correlation test (Rho), coefficient of determination ($R^2$) of a linear regression model, Lin's concordance correlation coefficient (CCC), and p-value of the Spearman correlation between observed (x-axis) and predicted (y-axis) outcomes in the training cohort. **c** The validity of the three RF models was tested in an independent cohort of 98 samples that did not contribute in any way to development of the models. Concordance line (x = y) is shown in black. Linear regression lines are shown in black with gray-shaded error band representing 95% confidence interval. All statistical tests were two-sided. Source data are provided as a Source Data file.

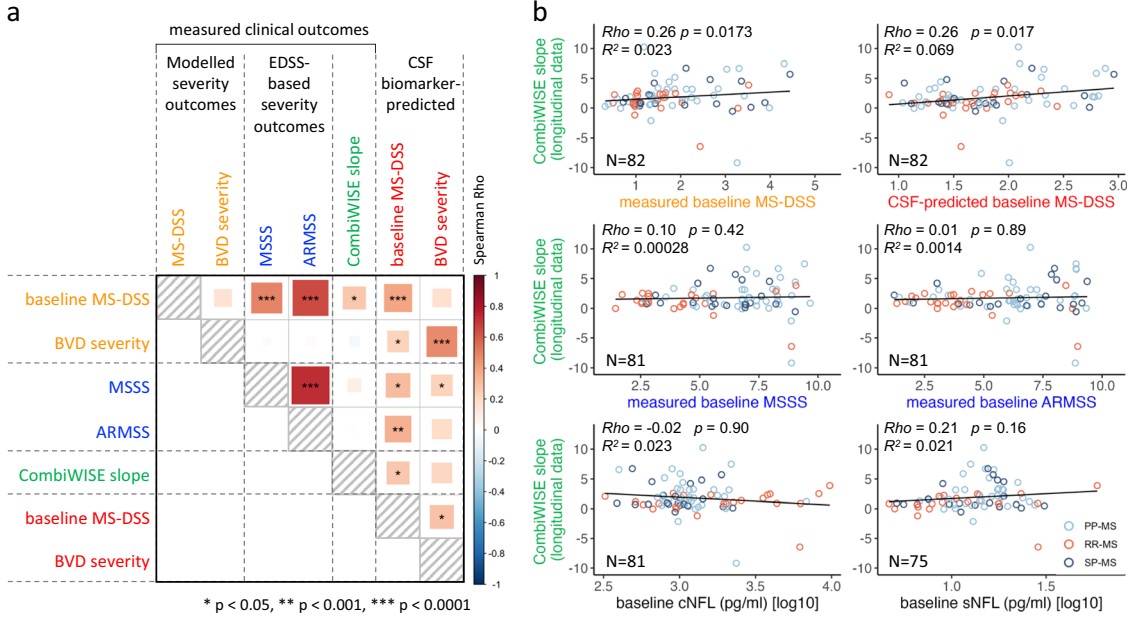

**Fig. 7 | MS severity outcomes in the validation cohort and their associations with cerebrospinal fluid (CSF) model predictions. a** Spearman correlation (the size and color of the square represent the Spearman Rho; significance levels are depicted by stars) between five measured clinical outcomes and two CSF biomarker-predicted multiple sclerosis (MS) severity outcomes in the validation cohort (N = 98). **b** Correlations between prospectively measured MS progression slopes (i.e., therapy adjusted CombiWISE slopes derived from longitudinal clinical

follow-up; y-axes), clinical/imaging outcomes and CSF biomarker-predicted outcomes (x-axes). For exact Spearman Rho, p-values, and $R^2$ see Supplementary Data 8. MS-DSS Multiple Sclerosis Disability Severity Score, MSSS Multiple Sclerosis Severity Score, ARMSS Age-Related Multiple Sclerosis Severity, CombiWISE Combinatorial weight-adjusted disability score, sNFL serum neurofilament light chain, cNFL CSF neurofilament light chain. All statistical tests were two-sided. Source data are provided as a Source Data file.

CSF biomarker-based models also predicted all EDSS-based MS severity outcomes with statistical significance and weak effect sizes (Rho 0.24–0.38; Supplementary Data 8).

Finally, we compared predictive effects of CSF-biomarker-based models with NFL measured in the CSF (cNFL) and serum (sNFL). Most NFL measurements were part of recently published paper[14], where we made unexpected observation that while cNFL strongly outperforms sNFL in predicting acute MS injury reflected by contrast-enhancing lesions (CEL) on brain MRI, only sNFL but not cNFL correlates (weakly) with MS severity outcomes. This sNFL advantage resides in its ability to capture spinal cord injury that leads to release of NFL into systemic circulation (likely from axons of spinal roots and peripheral nerves), bypassing the CSF[14]. However, while sNFL explains 5.7% variance of baseline MS-DSS (p = 0.023) neither cNFL nor sNFL predict future rates of disability accumulation (Fig. 7b and Fig S2).

Thus, we conclude that CSF biomarker-based models outperform NFL in predicting future rates of MS disability accumulation.

## SomaScan-based models of MS severity reveal pathophysiological heterogeneity among MS patients that transcends clinical classification of MS subtypes

We alluded to the possibility of heterogeneity in disease mechanisms that underlie MS severity, which is not captured by clinical MS severity

outcomes, but may be reflected in CSF biomarkers. To explore possibility of such pathogenic heterogeneity, we performed unsupervised cluster analysis[42,43] of MS patients using CSF proteins from the three MS severity models.

Seven distinct patient clusters (Fig. 8) differed in CSF concentrations of proteins from four protein modules: 1. Myeloid lineage/TNF module (Module 1; red annotation; Supplementary Data 9); 2. CNS repair module (Module 2; green annotation; Supplementary Data 10); 3. Complement/coagulation module (Module 3; blue annotation; Supplementary Data 11); and 4. Adaptive immunity and CNS stress module (Module 4; black annotation; Supplementary Data 12). The protein module names were based on STRING annotations (Supplementary Data 9–12).

All MS severity models selected biomarkers from all four modules (Fig. 8). While the MS clinical subtypes (i.e., RRMS, SPMS, and PPMS) were distributed across all seven molecular groups, few minor differences were noted: patient cluster 2 had a predominance of male progressive MS patients. This cluster had relatively low expression in the CNS repair module and high expression in the Myeloid lineage/TNF module and Complement/coagulation module. Consequently, these patients had higher MS severity. In contrast, patient clusters 3 and 4 were relatively enriched for female patients. Patient cluster 3 had only high expression of protein module 4 (Adaptive immunity and CNS

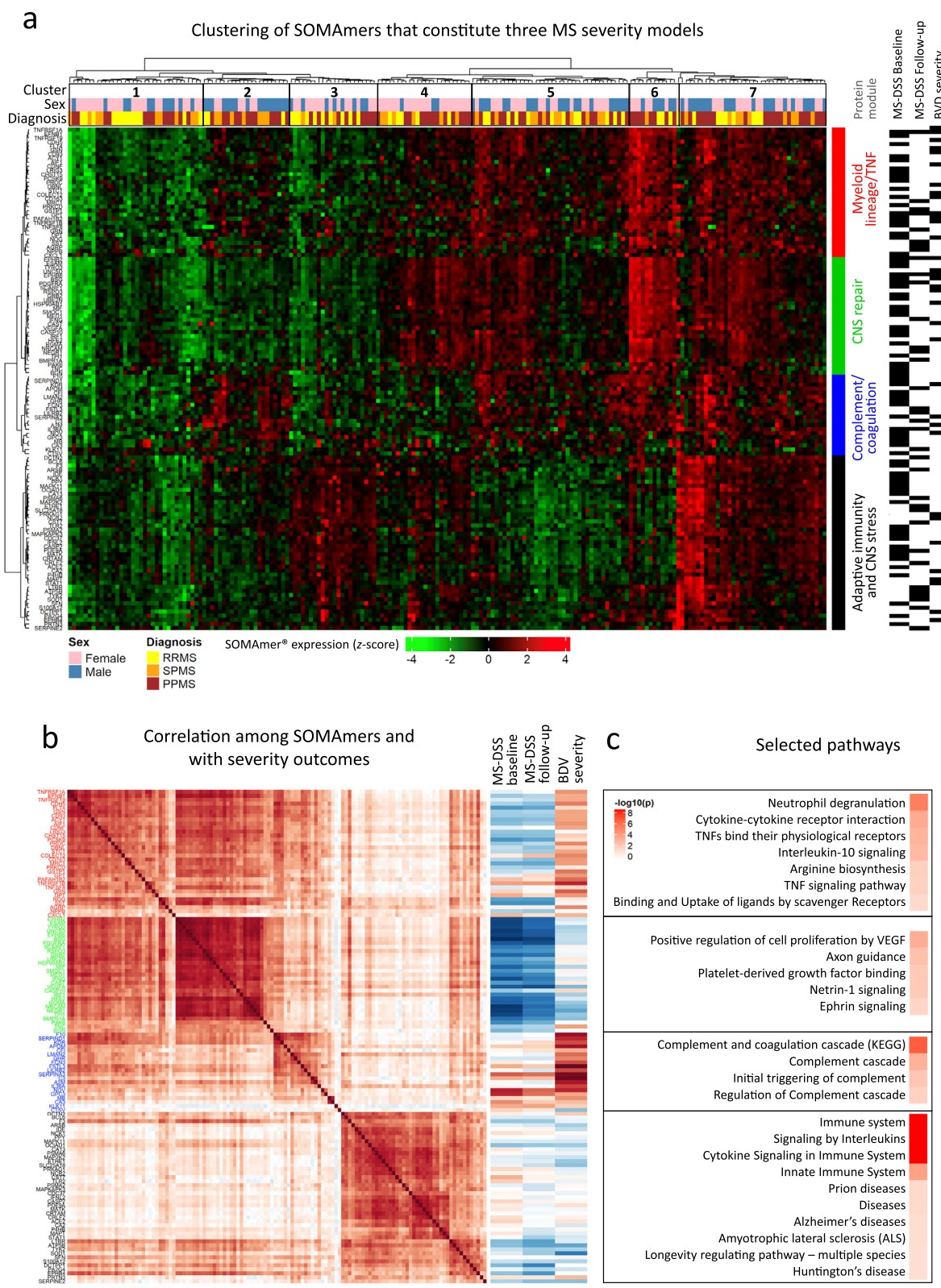

**a** Clustering of SOMAmers that constitute three MS severity models

**b** Correlation among SOMAmers and with severity outcomes

**c** Selected pathways

stress) and was enriched in RRMS subjects. Patient cluster 4 had relatively high expression of all protein modules except module 3 (Complement/coagulation module), which meant that these patients had relatively low MS severity.

These data support different representations of mechanisms associated with MS severity that go beyond physiological pathways of sexual dimorphism and may underlie differences in prognosis between male and female MS patients.

## Discussion

Developing treatments that inhibit disability progression require understanding of mechanism(s) that cause CNS tissue injury. However,

**Fig. 8 | SomaScan-based models of multiple sclerosis (MS) severity reveal pathophysiological heterogeneity among MS patients. a** Heatmap displaying the log-expression of the selected proteins from the three severity models in the MS cohort, with hierarchical cluster analysis identifying four protein modules (rows) across seven patient clusters (columns). RRMS relapsing-remitting multiple sclerosis, SPMS secondary progressive multiple sclerosis, PPMS primary progressive multiple sclerosis, MS-DSS Multiple Sclerosis Disease Severity Score, BVD brain volume deficit. Black rectangles on the right of the module annotations indicate whether the specific protein was present in a given model. **b** Spearman correlation plot of pipeline-selected biomarkers, ordered by module membership (left), along with Spearman correlation coefficients between model-selected biomarkers and measured MS severity outcomes (right). Colors of the protein labels correspond to module membership in **a**. **c** Selected pathways identified using STRING analysis, along with false-discovery rate (FDR)-adjusted $-\log_{10}$ $p$-values for the four protein modules, respectively. See also Supplementary Data 9–12. Ordered list of proteins displayed in the heatmap (8**a**) and correlation matrix (8**b**) is available in Supplementary Data 16. Source data are provided as a Source Data file.

identifying disease mechanism for polygenic CNS diseases is challenging because they occur behind the BBB, pathology studies can't differentiate causal processes from epiphenomena and disease mechanisms are inadequately reproduced in animal models. CSF biomarkers offer complementary information and provide ability to link intrathecal molecular processes to clinical outcomes. This study shows that CSF biomarkers can be aggregated to models that correlate with clinical and imaging MS severity outcomes and predict future rates of disability accumulation, measured by prospective longitudinal follow-up of MS patients in the independent validation cohort.

We will first address the study limitations: The cohorts are relatively small if judged by EDSS-based outcomes, raising concerns about statistical power. Statistical power is the probability with which we'll detect true relationship when the true relationship exists. Clearly, we detected (training cohort) and validated in new patients, relationships between all three CSF biomarker-based models and MS severity outcomes.

However, while in training cohort models explained >60% of variance, this decreased to 17–26% of variance explained in the independent validation cohort. It is tempting to think that using substantially larger cohorts would yield models with stronger validated effect sizes. However, while training models in much larger cohort would likely decrease model's overfit, effect sizes depend on the outcome: the accuracy with which it is measured and how homogeneous is the biology that underlies it. This is demonstrated in serum SomaScan-based models of 11 quantitative health outcomes: slight decrease of models' validation performance was seen even when using thousands of subjects in the training cohort, but validated effect sizes (or whether the model validated at all) were entire dependent on the outcome, not on the size of the training and validation cohorts[44]. There is substantial inaccuracy in MS severity measurements that stems from differences in performing neurological examination, translating neurological examination into a single number, but also in motivation and cooperation of the patients. This inaccuracy is reflected in modest correlations among MS severity outcomes. We believe that outcome inaccuracy determines the hierarchy with which outcomes correlated with CSF proteins (e.g., 13–76 times higher number of biomarkers correlated with MS-DSS than with EDSS-based outcomes) and predict longitudinally measured MS progression slopes (Supplementary Data 8).

Thus, the imprecision of measuring MS severity and the heterogeneity of the mechanisms that underlie it limit the effect sizes with which any model may predict MS severity. Consequently, our results are best interpreted in comparison to published literature. To do so, we recently published meta-analysis[45] of 302 publications that used clinical, imaging, or biomarker-based predictors of MS clinical outcomes: Table 2 of that meta-analysis summarizes studies predicting MS severity as continuous outcomes. The training cohorts' results explained maximum of 45% of variance, while independent validation cohorts explained maximum of 12% of variance. The meta-analysis also shows that decrease in effect sizes from training to validation cohorts is not an anomaly, but a rule. Furthermore, only 8% of publications validated effect sizes in new cohort. We conclude that CSF biomarker-based models of MS severity in current study achieved highest effect sizes in both training and validation cohorts. Validated effect sizes are more than two-fold higher than the strongest published validated model, using any type of predictor, including MRI.

Current models also outperform NFL, currently the most useful single biomarker of CNS injury. Increased NFL reliably identifies people with acute or subacute neuroaxonal injury such as subjects forming new MS lesions. While some (but not all) studies also linked NFL measurements to future MS progression, the published studies emphasized $p$-values rather than effect sizes[13,16,46], which are comparable to what we measured in the validation cohort here.

The advantage of CSF biomarker-based models over NFL resides not just in stronger prognostic power, but in their ability to reflect potential disease mechanisms, whereas NFL is an epiphenomenon of axonal injury. Indeed, the important biological insight learned from this study is the fundamental role CNS tissue plays in determining MS severity and that its influence dominates the MS severity measures based on physical disability, while coagulation and clotting cascades are stronger determinants of the BVD severity.

We also observed that, to the extent to which measured CSF biomarkers reflect physiological aging (which is 97% of variance for HV; Fig. 4b), MS is not associated with accelerated physiological aging on a molecular level. In fact, age-discordant CSF proteins (i.e., decreased in healthy aging and increased in MS aging; Fig. 2a) point towards re-expression of CNS developmental pathways related to axon guidance, EPHB2, EPHB4, EPHB6, NTN1, NOTCH1, NOTCH3, and SHH, which likely mediate CNS repair, as these proteins and their signaling pathways negatively correlate with MS severity.

NOTCH-related signaling was especially strongly and negatively associated with the rates of development of clinical disability. NOTCH-signaling pathways have overarching effects on many MS-related processes, including CNS repair (adult neurogenesis, formation of new synapses and remyelination)[47], neovascularization and vascular damage (especially NOTCH3), even the immune system[48].

This result has important implications: while the prevalent notion blames neurodegenerative mechanisms for disability progression in MS, our results identified lack of neuro-reparative processes, not only those linked to remyelination, but also those that directly affect neurons, as having validated CNS association with disability-based MS severity. Indeed, while these pathways decrease with natural aging, their re-expression in MS confers better prognosis. Thus, new research is needed to provide mechanistic insight, which could translate into treatments strengthening these physiological neuro-reparative mechanisms, that can be clearly re-expressed even in older progressive MS patients.

The dichotomy of molecular pathways associated with the rates of accumulation of physical disability versus with BVD severity is fascinating, as it may finally explain why some MS patients have severe brain damage on structural MRI imaging (i.e., large T2 lesion load and prominent brain atrophy), but they have surprisingly low physical disability; whereas other MS patients with minimal brain damage accumulate physical disability at high rate from disease onset (e.g., PPMS, especially male subjects).

We already mentioned that NOTCH signaling-related GO/REAC-TOME terms were strongly associated with both MS-DSS models, while coagulation and clotting cascades dominated BVD severity. Our findings expand mechanistic studies from animal models and human post-mortem studies that link vascular permeability, resulting in the influx of plasma proteins, such as fibrinogen and complement components to CNS tissue, with subsequent brain damage[49].

This finding poses an important question: why aren't the coagulation/platelet activation-related pathways equally associated with disability-based MS severity outcomes? Perhaps the explanation lies in the molecular differences between brain and spinal cord tissue, with the latter being the dominant anatomical site associated with clinical disability[50–52]. The beneficial CNS processes may also dominate, so that in their presence, the increased vascular permeability and influx of plasma proteins, including complement, does not cause neuronal or axonal damage.

Our results also inform on the long-standing question whether CNS tissue damage outside of MS relapses and especially at the progressive stage of MS is caused by compartmentalized inflammation or neurodegenerative mechanisms: on a group level, CSF biomarkers associate MS severity with both CNS- and immune-related pathways. From the immune-mediated mechanisms both this study and previous genetic studies[53,54] singled out immune effector mechanisms that cause cell death, such as cell-mediated cytotoxicity (i.e., cytotoxicity of T cells, but also NK cells and neutrophils and monocytes/macrophages), and complement-related processes as reproducibly associated with MS severity.

In this regard, a biased knowledgebase of public databases towards cancer biology with underrepresentation of CNS processes somewhat limits interpretation of these associations. For example, increased CSF levels of early complement proteins may not reflect their blood origin, but rather a proinflammatory, toxic response of microglia and astrocytes[10,55], even though this biology was not annotated in pathway analyses. Hence, mechanistic studies must follow our results to identify cellular sources of biomarkers assembled in CSF biomarker-based models and the conditions under which they are released and consumed during physiological and pathogenic interactions between CNS and immune cells.

Lastly, intra-individual heterogeneity in pathways linked to MS severity observed in this study is highly reminiscent of pathological heterogeneity involved in the formation of acute MS lesions[56]. This information is essential for development of new, process-specific treatments aiming to slow CNS tissue destruction in patients with residual progression on immunomodulatory drugs as it shows that approximately a half of MS patients lack any of the four mechanisms identified in this study. Thus, without CSF biomarker guidance, almost half of the participants in clinical trials of novel treatments may lack the target of the tested medication. This will dilute therapeutic response on a group level, requiring prohibitively large Phase 2/3 trials. Even if such an expensive drug development succeeds, the blind application of such treatments will incur high societal cost and unnecessarily expose patients who lack therapeutic targets to the side effects of applied drugs.

In conclusion, CSF analysis for oligoclonal bands was essential for MS diagnosis 40 years ago but was outpaced in contemporary practice by non-invasive CNS imaging. Advanced proteomic assays applied to CSF have a potential to revolutionize drug development and personalize treatments for MS and other CNS diseases[57]. We expect that the clinically useful information derived from CSF biomarkers will continue to expand and will eventually include predictive models to match therapy to the molecular mechanisms that drive disease process in individual patients. This will make treatments simultaneously more effective, safer, and cost-efficient.

## Methods

### Subjects
MS patients and HVs were prospectively recruited between May 2004 and April 2021 under an approved IRB protocol "Comprehensive Multimodal Analysis of Neuroimmunological Diseases of the Central Nervous System" (Clinicaltrials.gov identifier NCT00794352) and signed written informed consent (samples collected before 2009 were part of the "NIB Repository Protocol".

[10-N-0210]). To be considered for the study, patients must have had a clinically definitive MS diagnosis, a lumbar puncture (LP) within one year of a clinical visit that included four clinical scales (i.e., EDSS[17], Scripps Neurological Rating Scale (SNRS)[58], nine hole peg test (9-HPT), and 25 foot walk (25FW)), which are all required for calculation of CombiWISE[18].

To assure that CSF biomarkers, imaging, and clinical data were not influenced by treatments or MS exacerbations, patients were excluded if they were in MS exacerbation or have been on low-efficacy therapies (i.e., Copaxone, interferon-beta preparations, and oral DMTs) within 3 months of LP, or high-efficacy therapies (i.e., Natalizumab, Daclizumab, Alemtuzumab, Rituximab, or Ocrelizumab) within 6 months of LP [note that the classification of drugs into low and high efficacy was adopted from a meta-analysis of age-adjusted efficacies from controlled clinical trials[59]].

HV inclusion criteria were ages 18-75, lack of neurological diagnosis or systemic disease that could influence neurological disability or brain MRI, and with vital signs in the normal range during the initial screening. The demographic data of all subjects are detailed in Table 1.

### Clinical data
Patients underwent neurological examination by an MS-trained clinician. Before November 2017, the calculation of neurological rating scales EDSS and SNRS was performed by each clinician. After November 2017, the calculation of all neurological rating scales was fully automated using NeurEx™ App[24], which also computes the NeurEx™ score, a continuous disability score ranging from zero to theoretical maximum of 1349. For clinical visits linked to CSF collection before 11/2017, an MS-trained clinician retrospectively transcribed the neurological examination documented in NIH electronic medical records into NeurEx™ App. Clinicians rating neurological disability were blinded to volumetric MRI data and CSF biomarker data, as well as to calculated MS severity scales (described below).

Non-clinical investigators, blinded to neurological disability scales, MRI volumetric data, and CSF biomarker data collected 25FW and 9-HPT and uploaded these to the research database. All clinical and functional data were quality controlled during weekly clinical care meetings after which the corresponding parts of the database were locked to prevent modifications.

CombiWISE was automatically computed in the research database from EDSS, SNRS, 25FW, and 9-HPT values as described[18]. Machine learning-optimized MS-DSS was computed as described[21]. MS-DSS predicts future rates of disability progression as opposed to EDSS-based severity scales—MSSS[19] and ARMSS[20].

All computed scales developed by the Bielekova lab are freely available at https://bielekovalab.shinyapps.io/msdss/. NeurEx™ software is likewise freely available to non-commercial entities.

### CSF processing
CSF was collected on ice and processed according to a written standard operating procedure by investigators blinded to clinical and MRI outcomes. Aliquots were assigned alphanumeric identifiers and centrifuged for 10 min at 4 °C within 15 min of collection. Until use supernatant was aliquoted and stored in polypropylene tubes at −80 °C.

### SomaScan®
CSF samples were analyzed blindly, using SomaScan® technology[60] (Somalogic Inc, Boulder, CO, USA), a DNA aptamer-based assay that measures relative fluorescence units (RFUs) of 1,305 proteins (available after October 2016, referred to as the 1.3 K platform) by the NIH Center for Human Immunology. In total, 227 MS patients and 24 HVs (42 unique samples) had CSF samples available on the 1.3 K platform that met the inclusion criteria discussed above.

## Magnetic resonance imaging (MRI)-based MS severity scale

The brain MRIs were performed on 1.5 T and 3 T Signa units (General Electric, Milwaukee, WI) and 3 T Skyra (Siemens, Malvern, PA) equipped with standard 16- and 32-channel imaging coils.

MRI sequences used for grading comprised of T1 magnetization-prepared rapid gradient-echo (MPRAGE) or fast spoiled gradient-echo (FSPGR) and T2 weighted three-dimensional fluid attenuation inversion recovery (3D FLAIR). The details of the MRI sequences are previously published (32).

The brain MRI images were evaluated by two complementary methods: 1. semiquantitative ratings were assembled to Combinatorial MRI scale of CNS tissue destruction (COMRIS-CTD) using the published formula (32), available at https://bielekovalab.shinyapps.io/msdss/; 2. Identical MRI scans were analyzed using LesionTOADS volume segmentation algorithm[61], performed internally at the NIH until December 2018 and afterwards in collaboration with QMENTA platform (https://www.qmenta.com/).

Raw unprocessed but locally anonymized and encrypted T1-MPRAGE or T1-FSPGR and T2-3D FLAIR DICOM files as input sequences, ideally with 1 mm$^3$ isotropic resolution, were uploaded to the QMENTA platform. LesionTOADS, now implemented into the cloud-based service, is a fully automated segmentation algorithm using multichannel MRI data[62]. The uploaded sequences are anterior commissure-posterior commissure (ACPC) aligned, rigidly registered to each other and skull stripped (the T1 image is additionally bias-field corrected). The segmentation is performed by using an atlas-based technique combining a topological and statistical atlas resulting in computed volumes for each segmented tissue in mm$^3$. Manual quality control of the scans was performed to check for inaccurate segmentation of brain structures, low image quality, and motion artifacts.

To calculate the BVD severity measurement, brain volume deficit measured as 1-BPFr (calculated as proportion of intracranial volume occupied by brain tissue; [Cortical gray matter + Caudate + Thalamus + Putamen + Normal appearing white matter + Lesions]/[Cerebrum gray matter + Caudate + Thalamus + Putamen + Normal appearing white matter + Lesions + Sulcal CSF + Ventricular CSF]) was regressed against age using baseline data in the full cohort of patients with MS. This demonstrated strong evidence of increasing brain volume deficit over increasing age ($t_{128} = 4.41$, p-value=0.00002). The residuals from the resulting regression were then calculated. These residuals were used as the BVD severity outcome, where positive values are indicative of more CNS tissue destruction in a manner analogous to clinical measures of MS severity.

## Adjusting SOMAmers for differences in age and sex

As previous studies[1,63,64] have demonstrated associations between specific CSF proteins measured by SOMAscan and confounding factors age and sex in HVs, we sought to adjust protein levels in our MS patients to account for natural physiological differences due to age and sex. An initial list of SOMAmers were selected from published INTERVAL cohort examining serum proteins using SomaScan[1] where either age or sex associations were detected. The natural log of these SOMAmers were modeled using regression to test for age and/or sex difference in the 1.3 K platform in CSF samples from MS patients as well as HVs. SOMAmers with an association between age and/or sex with $p < 0.05$ in the MS cohort, and concordant directions between INTERVAL HV serum and HV CSF, were adjusted in the MS data using regression models derived from HV CSF samples.

## Examining individual associations between SOMAmers and disease outcomes

Individual Spearman correlations were computed between adjusted protein levels and MS severity endpoints. All p-values for individual SOMAmer correlations were adjusted for multiple comparisons using the FDR method[38]. See also Supplementary Data 6.

## Constructing a CSF-based severity model of MS using statistical learning

Random forest algorithm[39] using the ranger R package[65,66] was used in RStudio software version 1.1.463 (utilizing R version 3.6.1) to construct the CSF biomarker-based models of MS severity. For each platform, the CSF samples at the untreated baseline were used to predict MS-DSS at both the baseline visit and the most recent follow-up, and the BVD severity measure at baseline. All possible protein ratios were included in the modeling along with individual markers. The principle of random forest algorithm and rationale for using protein ratios has been explained[3].

Prior to model development, the available data were randomly split into training and validation cohorts, with 129 samples used as a training cohort and 98 samples retained only for model validation. To reduce number of ratios/markers based on predictive performance, a variation of the published procedure[40] was performed (Fig. 6a)[54]. Briefly, 10 random forests were run using the training cohort, and variable importance measures based on node impurity[41] were averaged together. The bottom 10% of variables, according to these average variable importance measures, were removed from the candidate set. This process was repeated until only three variables remained. The mean and standard deviation of the out-of-bag (OOB) error was graphically assessed to determine the final cut point for each model. This procedure was performed for SOMAmers adjusted for age and sex. For each instance, a final random forest model was constructed in the training cohort, using ntree = 40,000 and mtry = 3*√p, where $p$ is the number of available features. Biological interpretations of the selected proteins were explored using cluster analysis[42,43], STRING analysis[36], and g:Profiler analysis[37]. The raw data and code are available as Supplementary Data 13 and 14.

## Statistics

All statistical tests are two-sided. All correlations were calculated using Spearman correlation coefficients. Therefore, no assessment of linearity was performed. When determining if SOMAmers had physiological age and sex associations, t-statistics from multiple linear regression models were constructed. When comparing differences between CSF-predicted age and observed age between diagnoses pairwise comparisons using Wilcoxon signed-rank test and FDR adjustment for multiple comparisons were used.

The MS disease heterogeneity was analyzed in the MS cohort by unsupervised hierarchal clustering of z-score-transformed values of SOMAmers selected by the three MS severity models using the "ward.D2" clustering method and Euclidean clustering distance as part of the "ComplexHeatmap" R package[67].

## Study approval

All subjects were prospectively recruited under protocol "Comprehensive Multimodal Analysis of Neuroimmunological Diseases of the Central Nervous System" (Clinicaltrials.gov identifier NCT00794352) and signed written informed consent. The study was reviewed and approved by the Intramural Institutional Review Board at the National Institutes of Health. Healthy volunteers received financial compensation for their participation in the protocol.

## Reporting summary

Further information on research design is available in the Nature Portfolio Reporting Summary linked to this article.

# Data availability

All relevant raw data supporting key findings of this study are available within this article and its Supplementary Information. Source data are provided with this paper. Biological interpretation of the selected proteins was explored using public databases

STRING (https://string-db.org) and g:Profiler (https://biit.cs.ut.ee/gprofiler/gost). Source data are provided with this paper.

## Code availability
Custom R codes used for data analysis are available as Supplementary Data 14.

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

## Acknowledgements

This study was funded by the Intramural Research Program of the National Institute Allergy and Infectious Diseases (NIAID) of the National Institutes of Health (NIH). This work utilized the computational resources of the NIH HPC Biowulf cluster. The authors thank Brian Brown, NIH Library Writing Center, for manuscript editing assistance.

## Author contributions

Study concept and design: B.B.; data acquisition and analysis: B.B., C.B., P.K., M.V., M.G.; collection of clinical data: A.W., M.S.; drafting of the manuscript and figures: all authors.

## Funding

## Competing interests

The authors declare no competing interests.
