## [Peer Review File · Nature Communications]

Molecular models of multiple sclerosis severity identify heterogeneity of pathogenic mechanismsReviewers' comments:

Reviewer #1 (Remarks to the Author):

The authors used machine learning (ML), the authors modeled 1305 cerebrospinal fluid (CSF) biomarkers in a training dataset of untreated MS patients (N=129) in order to predict past and future speed of central-nervous-system damage and disability accumulation across MS phenotypes. They then used data derived from 24 healthy volunteers to differentiate natural aging from MS-related mechanisms and quantified gender effects on CSF proteome. Resulting models were validated in an independent longitudinal cohort of 64 MS patients and permitted to identify pathways related to disease mechanisms that the authors propose as novel therapeutic targets.

This is a complex but interesting study that attempts at demonstrating that CSF-based proteomics modeled with deep learning is better in predicting patients clinical evolution than other conventional measures such as MRI. The study is appropriately written however sometimes it becomes difficult to follow because of its great complexity: I strongly advise to better structure the results to improve readability. Methodologically, I have several major and minor concerns that I list below:

Major comments:

- can you structure the results by adding the initial hypotheses and then the obtained model performance(s) after correction for multiple comparisons? It would really facilitate the reader...
- how did the author define a "strong validation"? This currently ranges from R2 0.3 to 0.7 in the results section. Can you please state the applied criteria and consequently present your results?
- Results, page 5 "Unfortunately, because EDSS increases on average by 1 point every 10 years in MS natural history cohorts, MSSS and ARMSS do not predict future rates of disability progression in MS cohorts with less than 10 years of follow-up." I would moderate this statement as it is not true. Please, try to provide more detailed arguments about the choice of the clinical test.
- MRI: patients were scanned at both 1.5 and 3T. Were single patients followed up at the same field strength? The differences in the contrast-to-noise ratio may largely affect both lesion detection/segmentation and atrophy estimation.
- Was brain parenchymal fraction (BPF) considered as a measure of atrophy? Atrophy requires a longitudinal assessment of brain volume changes. Was this performed? How was BPF calculated? Was any quality check performed? Was any lesion filling performed?
- Lesion TOADS: it is a method that is known to require extensive manual correction. Was this performed?
- Using the residuals of a fitting between BPF and Age as a measure of MRI disease severity is kind of interesting but not usual: this can simply reflect technical differences between scans... can you use instead brain volume changes or n. of new/enlarged lesions or n. of Gd lesions? These are more traditional measures that will help us to better understand the impact of the results.
- please add a limitation section to the manuscript

Minor comments:

Please, reduce the Figure legends as currently, they are very long and redundant with the main text.

In summary, I recommend extensive revisions before the manuscript can be considered as a candidate for publication.

Cristina Granziera

Reviewer #2 (Remarks to the Author):

This is a well performed CSF proteomics paper, which reveals interesting biology and heterogeneity in MS that could be predictive for several aspects of disease progression in MS. The power may not

be very high for the amount of proteins and different statistical analyses performed, it is a substantive study, for CSF being available and including a large number of SP and PPMS patients. The authors tend to overstate their findings (ccc 0.66 being interpreted as excellent, words as unambiguous, introduction of age and sex correction as something very unique for the study), and several of their statements in the discussion are not well sustained. Also some aspects need more explanation for the readers to fully understand the findings.

I would rate some of the analyses, for example the cluster analyses, as exploratory, and it would be nice to phrase these as such, to keep the focus on the main findings.

Lines 92-95, I cannot follow the reasoning that correction for EDSS or MRI scores for disease duration transforms them into an outcome with higher probability to be causative. Even though such correction improves the intra-individual slopes, they are still cross-sectional measures, and thus they can be a consequence of the biology. This is a fundamental part of the assumptions of the authors to draw their conclusions, and should thus be explained better, or the statement of causal relation should be weakened. It appears from figure 1 that the authors did correlate baseline levels with future follow-up. This in itself increases the probability that a causative biological process is observed. Probably they wanted to express this in lines 92-95?

Further, in line 119, they state that they correlated the baseline CSF biomarker models against the last follow-up scores. Was there no correction for baseline clinical scores? This would make the models stronger, as now the relations could simply do you already existing baseline relations, and thus still not causative. It is better to look at change in cognitive scores.

Line 256/257: please repeat the names of the severity outcomes here, to help the reader keeping track.

Line 277, there appears a baseline correlation been performed. This was not mentioned as a strategy earlier, and results for other pathways for this cross sectional baseline analyses should also be mentioned, for comprehensiveness.

It is not clear to me what the difference is for the results if the analyses are done against or not against the genome background. In fact, I do not understand how the genome background is included in the statistical analyses. Were genotypic data available of the patients? Or is this some function that can be applied in STRING?

. Phrasings as in this sentence 'unambiguous positive associations' appears too subjective, for the results section. Adding numbers to support the statement might help.

. [SEP]line 330: a ccc of 0.66 in a scale from 0 to 1 does not appear to me as an excellent fit, but rather a qualification of good fit would be more appropriate. Validation with a ccc of 0.36 does not appear a very successful validation, and this is probably due to the low power.

Line 344-346: it would be very interesting if the validation improved by the enhanced model approach. This results should be made explicit.

Line 380-382: here it get lost what has been done. This seems to relate to the research question in the introduction, which I did not understand well there. Now I start to understand that the correlation coefficient between the CSF biomarkers and MS-DSS is taken to predict the future disability? Is that correct? More explanation on what has been done exactly and how we should interpret the outcome (correlation between an outcome and a baseline correlation factor?). Especially since this analyses is one of the innovative aspects of the current study, it is important to explain these results well.

Line 400 and further: it is unclear to me if here the total number of MS patients was included in the modelling, or only a subset, or divided again in a training and validation cohort. And why not in the models of the enhanced cohorts, which are supposed to be more stringent and thus the information more valuable?

I miss guidance in the results section which experiments address the causality question, presented in

the introduction.

It would be interesting to see how the current model compare to NfL, which has as advantage of being measureable in blood.

I cannot agree with the statement that only extreme values were included, when at least 50% of the values were included, as in the Barro study in Brain.

In the discussion the authors state that the power was sufficient to give reproducible results. However, the models developed in the training cohorts overfitted the predictions in the validation cohort. My first idea would be that this is a power issue causing this discrepancy. Even though for such a longitudinal study, it is already a nice number of patients that were included, but still the numbers are relatively low, especially in comparison of the numbers of proteins analyses and numbers of analyses being performed. Thus, this statement of a good power is not sustained by the current data.

Correlation between another similar method, Olink, should also be included (which was not supergood, unfortunately). This is a method that is more similar than ELISA or mass spect, being also a multiplexed protein binder assay.

Also, more explanation for the relative lay person should be provided what was done with sex and age. however, it is seen as scientific rigorous to correct for age and sex, and thus, this is not a I cannot understand that this is novel. But where I am mostly confused is how they corrected for age, as they did apply a correction for age in the regression analyses, isn't? So, then these factors can be seen as a confounder, similar as in other studies. Please explain to clarify.

Line 521-5222: here is probably missed the result showing comparison of the biological markers included in the different models. In the paragraph from line 320 on, I see mention of different numbers of aptamer ratios for each model, but not whether these were overlapping between the modes. This statement should be sustained better and the results elaborated.

Line 536, the elaboration of notch in the discussion is a novel result, in order words, such details are not elaborated well in the narrative of the results and should be.

The discussion of the clotting/coagulation mechanism is hampered by not stating whether this mechanism was positively or negatively affected, reflected in which specific proteins, and thus by some more granularity in the discussion, and similar to the notch discussion, more details in the results. And further more clues, e.g. from the literature, what this clotting mechanism may mean in CSF. Why are these clotting proteins present in CSF? What would their role be?

I see a discrepancy in the reasoning: the results seem to inform the compartmentalisation hypothesis in MS progression. Yet, further on in this paragraph the authors seem to state that we cannot draw any conclusions, due to the cancer bias of the pathway annotation databases.

Reviewer #3 (Remarks to the Author):

This is a very interesting manuscript investigating the CSF of patients with MS by proteomics using a DNA-aptamer-based technology. The aim is to identify biomarker or signatures of biomarkers that can predict disability progression and pathophysiological differences in order to design future personalized treatments for MS patients.

The platform used could potentially identify 1305 proteins. A machine learning approach was chosen to model CNS damage and disability progression.

The major findings are that MS was not associated with accelerated parameters for aging.

Furthermore, different pathways were identified in the functional enrichment analysis that correlated with disease severity and MRI atrophy, suggesting heterogeneity of the pathomechanisms between individual patients. Finally, they could discriminate between disease severity and future disability

progression.

This study is innovative and important for the possible development of individualized treatments. Most conclusions seem to be supported by the presented data. However, there are a few points that require attention:

- I fully agree with the authors that one has to control and correct for many variables in such a study and a precise clinical characterization is essential. The authors have put a great emphasis to correct for age-dependent markers and gender differences. However, there are several variables that are not clear if it was corrected for or how this was dealt with:

1. Had all patients positive OCB? Or were there differences between OCB pos. and neg. patients?
2. Were the data adjusted regarding the blood-CSF-barrier? Some MS patients may have a small increase in QAlb as measure for the blood-CSF-barrier. This may lead to more blood-derived proteins in the CSF and may thus skew the analysis.
3. Differences in IgG production in the CSF have been described as predictor for progression. Were the data corrected/correlated for QIgG?

- Were the HV really healthy volunteers? Why did they receive a LP?

- The quality control should be described in the methods. Were all the 1305 proteins reliably detected in the CSF? How were missing data/values below detection limit dealt with?

- All patients were untreated at the time of LP and high efficacy drugs within 6 month of LP were excluded. However, the immunological effects of Alemtuzumab and probably also Rituximab/Ocrelizumab, last for longer than 6 month.

- The patients were not during relapse at the time of LP, however, it is not stated how long after the last relapse the LP was performed. This may have a great influence on the dynamics of the CSF changes. This would also have implications on the interpretation of the results. It can be assumed that after a relapse the protein composition in the CSF changes dynamically. Thus, could it be that the heterogeneity is rather a continuum, e.g. first the myeloid or complement module, then the adaptive immunity module and then the repair module?

- The patients had a rather long disease duration, even in the RRMS cohort of 4.8 and 4.2 years, respectively. An earlier timepoint, e.g. at first diagnosis of MS, would be more important since at this time the treatment decision is often made and knowledge of prognosis would be helpful.

- Nothing is said about the treatment of the patients after LP. I would assume that many of the patients received a DMD. This of course has an effect on the development of future disability. Thus the analysis for future disability accumulation.

- Could the different MS forms (RRMS/SPMS/PPMS) be discriminated by the profile? This was not clear for me.

- The authors (correctly) criticize that NfL is only predictive on a group level. However, are their results predictive on a single patient basis? And have they made a comparison of their prediction model with NfL measurements?

- With regard to clinical progression it would be interesting to have also information cognition and not only on physical disability. This has in particular relevance for the cortical atrophy observed in the MRI.

- Inclusion of spinal cord MRI data would also be of interest. This may explain for the dichotomy of e.g. low brain damage with high physical disability.

- Have any of the patients also received a brain biopsy? Could the findings of the heterogeneous pathways be linked to the heterogeneity of the neuropathology, e.g. for complement? The

discrimination of type 1 and type 2 lesions according to Lucchinetti et al. from the CSF would be extremely helpful.

Reviewer #4 (Remarks to the Author):

This article proposes an approach to use CSF protein biomarkers and machine learning to provide insight into multiple sclerosis pathophysiology, and claims that the results can support individual clinical prognostication. Using the CSF samples of 129 untreated MS subjects (RRMS = 31, SPMS = 37, PPMS = 61) and 24 healthy individuals, the authors extracted 1305 biomarkers, performed adjustments to minimize the effects of physiological aging and sex, then trained three random forest machine learning models to find evidence of molecular pathways associated with MS severity and brain atrophy. Validation was performed on an independent data set from 64 MS subjects, with correlations to MS Disease Severity Score (at baseline and follow-up of mean 4.2 years) and "MRI Severity" (brain atrophy) as outcomes. These correlations characterized the predictive value of the biomarkers and were moderate, ranging from 0.4 to 0.6. One key biological insight was that the molecular pathways in determining MS severity may be different from the determinants of brain atrophy severity.

This paper has a number of strengths. Despite decades of research, MS remains a very difficult disease to predict, and novel techniques to further elucidate the molecular mechanisms of MS pathology and produce more accurate individualized prediction models are welcomed by the MS community. The study appears to be carefully conducted and the manuscript is clearly written. The selection of methods seems to be sound, including the use of the MS-DSS (recognizing the limitations of the EDSS for short-term prognostication), the overall proteomics framework, and the random forests with iterative variable selection. The biological insights on the molecular pathways governing MS severity and brain atrophy appear to be significant and novel (although I am not an expert in this area).

However, there are also a number of significant weaknesses. The number of subjects used for this study is simply too small to make any claims on the results providing "reliable prognostic information". At best this is a proof-of-concept study. The authors partially acknowledge this by stating "The sample sizes of the training and validation cohorts may seem small if judged by conventional, EDSS-based outcomes.", but MS is such a heterogeneous disease that it is imperative that the study cohort is biologically and clinically representative of the general MS population, and there is no statistical evidence that this is true. Another problem is that some key methodological details are missing, particularly for the machine learning components. For example, no description of the unsupervised clustering algorithm mentioned on p. 22 was provided, and the procedure for determining the hyperparameters of the random forest models was also missing.

Another weakness is that the manuscript appears to downplay the importance of MRI for MS prognostication. The authors used normalized whole brain volume to represent "MRI severity", but whole brain volume represents a small fraction of the information that can be derived from imaging, especially with quantitative MRI such as magnetization transfer or myelin imaging, with the former being widely available. Also, it is well known that cross-sectional brain volume measurements, such as the brain parenchymal fraction employed in this study, are not optimal for longitudinal studies, and registration-based methods such as FSL SIENA are much more precise and accurate for determining change. The authors suggest that CSF testing to estimate age-adjusted brain volume can obviate the need for actual MRI measurement, which is not well-founded given that image-based measurement has already established its scientific validity, practicality, and clinical utility.

Recommendations:

This manuscript has merit for publication, with significant modifications. The focus should be on the biological insights discovered, and claims of supporting precision medicine should be removed until a

much larger study can be performed. "MRI severity" should be renamed "brain atrophy severity" and the narrow view of imaging in this study should be acknowledged. More details on the machine learning methods should be provided, so that the experiment can be potentially reproduced.

Minor issues:

- Use "sex" instead of "gender"; this is the current scientific standard.
- Page 7, "NeurEx-based outcomes are more accurate"; the statement is about variability, so perhaps "accurate" should be replaced by "precise".
- Page 17, "training cohort performance grossly overestimated performance in the new set of subjects"; In machine learning, training performance simply estimates fit of the model to the training data, and does not estimate performance in new data.
- Page 20, "correlations of MRI severity... outcomes is only weak" should be "correlations of MRI severity... outcomes are only weak".
- Page 26, There is a strange link:
(<http://www.keystonesymposia.org/index.cfm?e=web.Meeting.Program&meetingid=1654>)

Point-by-point responses to the reviewers

Dear reviewers,

We thank you for your time in reviewing our manuscript. The COVID19 pandemic and subsequent closure of the NIH clinical center led to almost 2-year restriction of research activities. This understandably slowed our progress in analyzing more patients, generating and analyzing the requested NFL data and resubmitting the manuscript. Based on the editor's guidance we have generated CSF biomarker data on all MS patients that had LPs and sufficient follow-up data to fulfill inclusion criteria of this study and thus we expanded the validation cohort. We are happy to report that this expansion did not impact the results of the study.

There are few general themes we would like to address: We appreciate that the effect sizes from our independent validation cohort appear modest. Understandably, editors, reviewers and readers compare effect sizes across studies with different attributes of study design. However, attributes of the study design have defining implication on generalizability of study results¹⁻⁶. To prove this point and to facilitate comparisons of MS modeling studies we performed meta-analysis of published models of MS clinical outcomes (n=302 publications), not only compiling the reported effects sizes, but also scoring technical quality of study design⁷.

Consistent with publications from non-MS fields¹⁻⁶, this meta-analysis shows that quality of study design dramatically affects effect sizes: the poorer the study design, the higher effect sizes studies report. Out of 7 study design criteria that prevent bias and enhance reproducibility of the results, published MS studies fulfill on average 1 and no study fulfilled more than four. Only 8% of published studies used independent validation cohort. The reviewers and the editor considered our study small; but compared to published literature, our study is one of the largest reported in MS and fulfills 7/7 study design criteria. Based on these objective comparisons, the current study represents the most rigorous study design for successfully modeling and validating MS severity outcomes.

The reviewers attributed decrease in model's performance from training to independent validation cohort to "low power". Power depends on effect size and sample size, but power exerts no influence on validated effect sizes. All 3 CSF-biomarker-based models of MS severity achieved very low p-values in the independent validation cohort, proving that we had sufficient "power". Decreases in effect sizes, from training cohort to training cohort cross-validation and then to independent cohort validation are seen in virtually all machine-learning studies that report such comparisons⁶ (which in MS field are very few⁷). Yes, it is likely that a study that included thousands patients would derive models with lower degree of overfit⁶, but not necessarily with stronger validation effect sizes.

Validated effect sizes depend on the amount of noise. Variance of CSF biomarkers measured by DNA aptamers is about 10%⁸, but much higher in measuring brain atrophy⁹ and MS severity (see data added to this paper). We cannot "measure" MS progression or MS severity the same way we measure distance between 2 points on a sheet of paper: the latter

is fully defined and is measurable with high precision irrespective of measuring tool or measuring units. In contrast, disability is a concept, physical substrate of which is not only the number of neurons, but their connections and the environment in which they operate. It is unrealistic to expect much higher validated effect sizes for MS severity than what we observed in this study. To support this conclusion, the Table 2 of the cited meta-analysis of published MS models⁷ shows that using ANY type of predictors (i.e., MRI biomarkers, serum or CSF biomarkers including NFL, or genes) the maximum achieved effect sizes in predicting MS severity outcomes were 45% of variance explained in the training cohort and 12% of variance explained in the independent validation cohorts. We achieved max 63% variance explained in the training cohort and max 26% of variance explained in the validation cohort. In conclusion, current paper achieved the highest validated effect sizes in predicting MS severity outcomes. We have now added entire new section to the results (lines 405-454) and discussion (lines 513-542) dedicated to this comparison and explanation of effect sizes.

However, it would be a mistake to measure the value of this study only by validated effect sizes: if we measure MS severity using clinical, imaging outcomes or NFL, it tells us nothing about biological processes that operate in the intrathecal compartment of individual patients. CSF-biomarker-based models are useful not only because they predict future rates of disability progression, but because they measure potentially causal processes in living people. This allows screening of new drugs in Phase II clinical trials based on their ability to inhibit such potentially causal processes.

Finally, it is tempting to request more data when reviewing a paper, because no story is final. But equity should be part of the scientific review process: acquisition and analysis of clinical, imaging and biomarker data we included in this paper represents cost of several million USD. None of the 302 published studies we reviewed included such multi-modality measurements⁷. The reviewers asked for measuring the same CSF biomarkers using different technology (OLINK) or adding imaging outcomes not collected as part of this study (spinal cord volumes). This is equivalent to asking imaging study to reimagine the same organ using different scanner or asking imaging study to show that MRI-based models outperform models from CSF biomarkers. We hope that the additional patients/experiments we have provided and the analyses we performed to put validated results in the context of published literature will help the reviewers to objectively determine the value of current publication.

Reviewers' comments:

Reviewer #1 (Remarks to the Author):

The authors used machine learning (ML), the authors modeled 1305 cerebrospinal fluid (CSF) biomarkers in a training dataset of untreated MS patients (N=129) in order to predict past and future speed of central-nervous-system damage and disability accumulation across MS phenotypes. They then used data derived from 24 healthy volunteers to differentiate natural aging from MS-related mechanisms and quantified gender effects on CSF proteome. Resulting models were validated in an independent longitudinal cohort of 64 MS patients and permitted to identify pathways related to disease mechanisms that the authors propose as novel therapeutic targets.

This is a complex but interesting study that attempts at demonstrating that CSF-based proteomics modeled with deep learning is better in predicting patients clinical evolution than other conventional measures such as MRI. The study is appropriately written however sometimes it becomes difficult to follow because of its great complexity: I strongly advise to better structure the results to improve readability.

- We have extensively rewritten the paper

Methodologically, I have several major and minor concerns that I list below:

Major comments:

- can you structure the results by adding the initial hypotheses and then the obtained model performance(s) after correction for multiple comparisons? It would really facilitate the reader...

- We have added hypotheses or rationale for all result sections

- how did the author define a “strong validation”? This currently ranges from R2 0.3 to 0.7 in the results section. Can you please state the applied criteria and consequently present your results?

- This has been addressed in general comments above and extensively in new results and discussion sections.

- Results, page 5 “Unfortunately, because EDSS increases on average by 1 point every 10 years in MS natural history cohorts, MSSS and ARMSS do not predict future rates of disability progression in MS cohorts with less than 10 years of follow-up.” I would moderate this statement as it is not true. Please, try to provide more detailed arguments about the choice of the clinical test.

- We are not aware of any published literature that would invalidate this statement: a large observational study of 17,356 MS patients reported mean annualized EDSS

change of 0.08.¹⁰ (or less than 1 EDSS point every 10 years). Analogous progression rate was measured in placebo arms of MS clinical trials: e.g. In Ocrelizumab in PPMS clinical trial (ORATORIO) 39.3% of MS patients in placebo arm progressed on EDSS over 4.1 years of trial duration. And EDSS progression was defined as 1 point progression for patients with baseline EDSS \leq 5.5, and 0.5-point progression for patients with EDSS $>$ 5.5 – together this also leads to mean annualized EDSS change below 0.01 (or less than 1 EDSS point every 10 years).

- We now added these citations in support of our statement

- MRI: patients were scanned at both 1.5 and 3T. Were single patients followed up at the same field strength? The differences in the contrast-to-noise ratio may largely affect both lesion detection/segmentation and atrophy estimation.

- This is a cross-sectional MRI cohort, we do not show longitudinal data. Patients were scanned both on 1.5T (29.5% in the training cohort and 25.5% in the validation cohort) and 3T (70.5% in the training cohort and 74.5% in the validation cohort)

- Was brain parenchymal fraction (BPF) considered as a measure of atrophy? Atrophy requires a longitudinal assessment of brain volume changes. Was this performed? How was BPF calculated? Was any quality check performed? Was any lesion filling performed?

- Brain Parenchymal fraction was calculated as proportion of intracranial volume occupied by brain tissue; [Cortical gray matter + Caudate + Thalamus + Putamen + Normal appearing white matter + Lesions]/(Cerebrum gray matter + Caudate + Thalamus + Putamen + Normal appearing white matter + Lesions + Sulcal CSF + Ventricular CSF)]. Manual quality control of the scans was performed to check for inaccurate segmentation of brain structures, low image quality, and motion artifacts. This was now added into the manuscript Methods
- We respectfully disagree with the statement that brain atrophy cannot be determined from cross-sectional scans and requires longitudinal follow up: there are literally hundreds of publications from different CNS diseases that use BPFr cross-sectionally as measure of brain atrophy – e.g. to compare brain volumes in patients versus healthy volunteers or to compare correlations between BPFr and disability outcomes.
- Longitudinal studies are required to measure the RATES of brain tissue loss. But analogously to measuring clinical MS severity outcomes, the rate of CNS tissue loss can also be estimated from the cross-sectional study by relating BPFr to age: people with neurodegenerative diseases that affect brain have lower age-adjusted BPFr compared to healthy people. In this study we use this cross-sectional measurement of loss of brain tissue with age. We now made these cross-sectional calculations of MS severity clearer in our writing (e.g., see lines 152-158)

- Lesion TOADS: it is a method that is known to require extensive manual correction. Was this performed?

- Lesion-TOADS algorithm has been implemented into QMENTA platform and runs automatically. Posthoc quality control of the volumetric data was performed as described before.

- Using the residuals of a fitting between BPF and Age as a measure of MRI disease severity is kind of interesting but not usual: this can simply reflect technical differences between scans... can you use instead brain volume changes or n. of new/enlarged lesions or n. of Gd lesions? These are more traditional measures that will help us to better understand the impact of the results.

- We cannot use changes in new/enlarged lesions or number of (Gd) contrast-enhancing lesions (CEL) because these outcomes do not correlate with MS severity. CELs do not predict future rates of disability progression. This has been observed in numerous publications, nicely summarized in 1999 meta-analysis in the Lancet¹¹.

- please add a limitation section to the manuscript

- Limitations of the study have been already part of the discussion.

Minor comments:

Please, reduce the Figure legends as currently, they are very long and redundant with the main text.

- We fully agree with this comment and we greatly reduced figure legends

In summary, I recommend extensive revisions before the manuscript can be considered as a candidate for publication.

Cristina Granziera

We thank Prof. Granziera for taking responsibility for her review by signing it: this level of transparency is rare but extremely necessary. We deeply appreciate this step.

Reviewer #2 (Remarks to the Author):

This is a well performed CSF proteomics paper, which reveals interesting biology and heterogeneity in MS that could be predictive for several aspects of disease progression in MS. The power may not be very high for the amount of proteins and different statistical analyses performed, it is a substantive study, for CSF being available and including a large number of SP and PPMS patients. The authors tend to overstate their findings (ccc 0.66 being interpreted as excellent, words as unambiguous, introduction of age and sex correction as something very unique for the study), and several of their statements in the discussion are not well sustained. Also some aspects need more explanation for the readers to fully understand the findings.

I would rate some of the analyses, for example the cluster analyses, as exploratory, and it would be nice to phrase these as such, to keep the focus on the main findings.

- We carefully edited the manuscript to delete words such as unambiguous and added “exploratory” to cluster analysis
- However, we respectfully disagree with the implication of reviewer’s comment that the adjustment for physiological age and sex effects is not “unusual”. Even pure adjustment for age and sex as confounders is done only in 25% of published MS models⁷, while including healthy donor data to adjust for physiological age and sex effects on CSF biomarkers are almost never performed. The example of GDF15 in the Figure 2 explains the difference between the two methods.

Lines 92-95, I cannot follow the reasoning that correction for EDSS or MRI scores for disease duration transforms them into an outcome with higher probability to be causative. Even though such correction improves the intra-individual slopes, they are still cross-sectional measures, and thus they can be a consequence of the biology. This is a fundamental part of the assumptions of the authors to draw their conclusions, and should thus be explained better, or the statement of causal relation should be weakened. It appears from figure 1 that the authors did correlate baseline levels with future follow-up. This in itself increases the probability that a causative biological process is observed. Probably they wanted to express this in lines 92-95?

- Indeed, the reviewer is right that this is a fundamental part of the paper and if the reviewer did not understand it than we must have not explained it well. We have now rewritten/expanded that part of the paper: lines 82-125

Further, in line 119, they state that they correlated the baseline CSF biomarker models against the last follow-up scores. Was there no correction for baseline clinical scores? This would make the models stronger, as now the relations could simply do you already existing baseline relations, and thus still not causative. It is better to look at change in cognitive scores.

- We consider it likely that the reviewer is confusing MS progression (EDSS, CombiWISE, BPFr) scores with MS severity scores (MS-DSS, ARMSS, MSSS).

Correcting for baseline progression scores would indeed be useful to assess interim progression rates when comparing first and last clinic visits. However, such correction is not useful for MS severity scales, because they have time denominator: whatever progression of disability occurred between first and last clinic visit is already adjusted in calculation of MS-DSS, MSSS and ARMSS. Consequently, MS severity scores should be relatively stable characteristic of patients and we have now formally documented this using Interclass Correlation Coefficient (ICC) for MS-DSS (see figure S1).

- The main difference between primary (MS-DSS calculated from clinic visit concomitant with LP) and secondary outcome (MS-DSS calculated from last visit) is the way we measured disability (i.e., EDSS and CombiWISE): baseline MS-DSS was calculated from score generated from documented neuroexam in the electronic medical records, while majority of the follow-up severity scores were generated from neurological exams electronically documented using NeurEx™ App, which automatically computed disability scales, eliminating noise from translating examination into scale by different clinicians. We would also like to remind the reviewer that we use MS-DSS at last visit only as secondary outcome, as type of sensitivity analysis.
- Unfortunately, MS cognitive outcomes are not suitable for our analyses, because PASAT and even SDMT have high degree of “training effect” and high noise (i.e., variance in repeated measurements¹²), which is why these outcomes fail to show disease progression during MS clinical trials.

Line 256/257: please repeat the names of the severity outcomes here, to help the reader keeping track.

- We added names of the severity outcomes (now lines 292/293)

Line 277, there appears a baseline correlation been performed. This was not mentioned as a strategy earlier, and results for other pathways for this cross sectional baseline analyses should also be mentioned, for comprehensiveness.

- Unfortunately, we do not understand what the reviewer means: we only measured CSF biomarkers at one time (at the first clinic visit when patients were untreated), so the pathway analysis was performed from this first (untreated) visit data. But we see that this referred sentence is confusing and not adding much value, so for the clarity we deleted it.

It is not clear to me what the difference is for the results if the analyses are done against or not against the genome background. In fact, I do not understand how the genome

background is included in the statistical analyses. Were genotypic data available of the patients? Or is this some function that can be applied in STRING?

- To clarify, STRING was originally developed for transcriptomics data (such as RNAseq) and even though the database now expanded to cover proteins as well, by “default” it assumes that the sampling is done from the whole “genome” (in this case from the protein-coding sequences of the whole genome; meaning that the algorithm assumes we measured all ~20,000 human proteins, which is incorrect). This has an implication for calculating statistical significance: in a hypothetical example of using “whole genome background” the algorithm may calculate significant enrichment of cytokines with MS severity, if cytokines represent only 5% of human proteome but 20% of cytokines were associated with MS severity. But if we measured only ~1,300 proteins and cytokines represent 20% of those measured proteins, the algorithm will no longer mark association of cytokines as significantly enriched with MS severity, if we use correct “background” of only ~1,300 measured proteins.
- To avoid confusion, we have renamed the “genotype” to “proteome” in Figure 5

. Phrasings as in this sentence 'unambiguous positive associations' appears too subjective, for the results section. Adding numbers to support the statement might help.

- We removed the word “unambiguous” form the sentence

. [SEP]line 330: a ccc of 0.66 in a scale from 0 to 1 does not appear to me as an excellent fit, but rather a qualification of good fit would be more appropriate. Validation with a ccc of 0.36 does not appear a very successful validation, and this is probably due to the low power.

- We have addressed dichotomy between “power” and “effect sizes” in the general write up preceding the point-by-point rebuttal. In short: we validated all models with very low p-value, demonstrating that our study was not underpowered. Validated effect size is not related to power but to the accuracy of measurements (i.e., noise) and to true relationships between measured predictors and outcomes. We now expanded both Results and Discussion section to put our results into prospective of 302 published studies.

Line 344-346: it would be very interesting if the validation improved by the enhanced model approach. This results should be made explicit.

- The pipeline we used decreases model overfit by assuring that model contains only those CSF biomarkers that are contributing to correct predictions of all samples in the training cohort. By not “seeing” any samples from the independent validation cohort, the pipeline cannot eliminate all overfit. Already published papers, including our paper on CSF-biomarker-based diagnostic classifier of MS⁸ showed that employing such pipelines does improve validation.

Line 380-382: here it get lost what has been done. This seems to relate to the research question in the introduction, which I did not understand well there. Now I start the understand that the correlation coefficient between the CSF biomarkers and MS-DSS is taken to predict the future disability? Is that correct? More explanation on what has been done exactly and how we should interpret the outcome (correlation between an outcome and a baseline correlation factor?). Especially since this analyses is one of the innovative aspects of the current study, it is important to explain these results well.

- That is correct. The study design is shown in Figure 1: The primary outcome is MS-DSS measured concomitantly with the LP. This is the cross-sectionally measured MS disability outcome; effectively reflecting amount of accumulated disability at the time of LP with patient age (MS-DSS is much more complex model, also adjusted for several confounders and even for amount of CNS tissue damage seen on MRI). Like other (EDSS-based) cross-sectional MS severity outcomes (i.e., ARMSS and MSSS), these outcomes estimate rate of disability progression looking backward in time. But of course, clinicians and patients want to estimate FUTURE rates of disability progression. From 3 MS severity outcomes, only MS-DSS can do that¹³. To the extent to which CSF biomarkers capture information that has prognostic value, to that degree CSF-biomarker-based models may also predict future rates of MS disability progression. These FUTURE rates of disability progression were measured longitudinally, using CombiWISE scale measured at all clinic visits. We showed that CSF-biomarker-based model could predict FUTURE rates of MS disability progression in the independent validation cohort.
- We have now rewritten the paper to further clarify this important point.

Line 400 and further: it is unclear to me if here the total number of MS patients was included in the modelling, or only a subset, or divided again in a training and validation cohort. And why not in the models of the enhanced cohorts, which are supposed to be more stringent and thus the information more valuable?

- The clustering was done in the whole cohort, and we have clarified that now in the text and figure legend. However, this time we are not “modeling” anything, so there is no reason for splitting patients into training and validation cohort. Instead, we are using unsupervised clustering algorithm to

cluster all proteins that were selected by any of the 3 final (optimized) CSF-biomarker-based MS severity models. The algorithm clustered these proteins into 4 clusters which we then “named” based on their annotated functions. The same algorithm also clustered MS patients into 7 clusters, that differed in the measured relative concentrations of these 4 protein clusters: e.g., patients clustered into clusters 1&3 had low concentrations of proteins from complement and coagulation cluster. This means that if somebody decided to study eculizumab (antibody against C5a) in MS, close to 40% of MS patients would be unlikely to respond to this drug, because these patients do not have evidence of activated complement in their CSF. The fascinating observation is that even though the proteins were clustered algorithmically (i.e., without knowledge of protein functions), the proteins that were elevated together had also related functions.

I miss guidance in the results section which experiments address the causality question, presented in the introduction.

- The whole study is fully focused on identifying CSF proteins that can be assembled into models that predict MS severity outcomes, including future rates of MS disability accumulation. Everything that we found and discussed is candidate pathogenic (or beneficial) mechanism of MS, because it differentiates people who are progressing faster from people who are progressing slower in all age categories: in young patients as well as old patients. If these processes/pathways we identified were caused by MS, they should be underrepresented in young people with low levels of MS disability (RRMS) and overrepresented in old people with high levels of disability (SPMS, PPMS), but that is NOT what we have seen in Figure 8.

It would be interesting to see how the current model compare to NfL, which has as advantage of being measureable in blood.

- We measured NFL levels in the serum and the predictive power of serum NFL is greatly outperformed by our models (i.e., by $\geq 70\%$ for baseline MS-DSS, by $\geq 209\%$ for follow-up MS-DSS and by $\geq 633\%$ for MRI severity based on R^2):

- Please note that we have now published paper focusing on NFL that includes the data from this cohort¹⁴.
- The weak effect size of sNFL to predict MS progression is further supported by recent study that evaluated the ability of sNFL to predict disability progression in two placebo-controlled Phase 3 clinical trials of progressive MS: EXPAND trial that evaluated efficacy of siponimod in SPMS and INFORMS trial that evaluated efficacy of fingolimod in PPMS (4). This study reported that sNFL (the authors measured plasma NFL, which is comparable to sNFL) dichotomized to “low” (< than 30 pg/ml) and “high” (≥ 30 pg/ml) has significant predictive value to identify patients at risk of MS progression. Unfortunately, this study did not report AUC, sensitivity/specificity and predictive values of dichotomized sNFL, which are necessary to assess clinical utility of a test. Patients with CELs constituted 21.6% of EXPAND and 10.1% of INFORMS participants and in both trials, presence of CELs was the strongest predictor of elevated sNFL. Nevertheless, the subgroup analyses demonstrated that even patients without CELs had statistically significant increase in the risk of disability progression if they had high sNFL. But this study analyzed thousands of sNFL samples and achieved marginal p-values for subjects without CELs (e.g., $p=0.0274$ for $n=1147$ predicting 3 months confirmed disability progression in EXPAND trial). Because both effect size and number of subjects contribute to p-value, the marginal p-value can be

obtained from such large cohorts only if the effect size is very low. Thus, these data, generated from the gold-standard clinical trials support our conclusion that while sNFL correlates with MS severity on a group level, its accuracy is likely too low to be clinically meaningful on a patient level.

I cannot agree with the statement that only extreme values were included, when at least 50% of the values were included, as in the Barro study in Brain.

- For predicting EDSS worsening, Barro's study dichotomized sNFL by HD-based percentiles, using 90%-tile as primary outcome (Barro's Table 1; 432/1401 [30.8%] patients had NFL levels higher than 90%-tile). They also report marginal p-value ($p=0.021$; no adjustment for multiple comparisons is noted in Figure legend, results or methods section which suggest that after adjustment this difference would no longer be significant) for patient who had NFL in 80%-tile (642/1401 [45.8%]). So, 80%-tile cut-off excluded 54.2% of patients, which makes our statement: "The existing tools, such as NFL levels and BREMSO (the Bayesian Risk Estimate for MS at Onset), validated prognostic power by comparing only subjects with extreme values (i.e., excluding 50-90% of subjects)" valid.
- But the biggest difference between ours and Barro's BRAIN study is that Barro's study had 4.3% CIS and 69.3% RRMS patients: in other words, 73.6% of patients were able to form CELs, which is the type of biology where NFL shines: CELs increase NFL values above 90%-tile of (age-adjusted) HD values (see Barro's Table 2: NFL has strongest association with CEL). In Barro's study most patients who progressed on EDSS in subsequent 1 year progressed due to relapse. Our study did not dichotomize patients; we show that CSF biomarkers correlate with linear disability progression measured over >4 years.

In the discussion the authors state that the power was sufficient to give reproducible results. However, the models developed in the training cohorts overfitted the predictions in the validation cohort. My first idea would be that this is a power issue causing this discrepancy. Even though for such a longitudinal study, it is already a nice number of patients that were included, but still the numbers are relatively low, especially in comparison of the numbers of proteins analyses and numbers of analyses being performed. Thus, this statement of a good power is not sustained by the current data.

- We reiterate what we stated in general comments: power is reflected by the fact that we validated all 3 models with low p-values. Validation effect size has nothing to do with power and everything to do with how accurate we can measure MS severity and how MS severity clinical or imaging outcomes reflect uniform biology (see edited Discussion for more).
- Please note, that in the sNFL study in two Phase 3 MS trials we referred to above, the p-value for the ability of sNFL to predict MS progression in

n=1147 subjects was much weaker than what we achieved in much smaller independent validation cohort.

Correlation between another similar method, Olink, should also be included (which was not supergood, unfortunately). This is a method that is more similar than ELISA or mass spect, being also a multiplexed protein binder assay.

- Somascan assay is a highly reproducible assay and we have demonstrated it by extensive analysis of longitudinal and technical replicates, including some comparisons of Somascan vs ELISA-measured biomarkers⁸. Therefore, we do not see the rationale for performing Olink, especially as the reviewer correctly stated that the accuracy of Olink is lower than of Somascan

Also, more explanation for the relative lay person should be provided what was done with sex and age. However, it is seen as scientific rigorous to correct for age and sex, and thus, this is not a I cannot understand that this is novel. But where I am mostly confused is how they corrected for age, as they did apply a correction for age in the regression analyses, isn't? So, then these factors can be seen as a confounder, similar as in other studies. Please explain to clarify.

- First, adjustment for covariates is scientifically rigorous but rarely (25% of MS published studies) performed⁷. Especially for some MRI brain volumetric outcomes, these confounders can explain as much as 60% of variance. The reason why most MS studies omit adjustment for covariates is that age itself is also a strong predictor of disability or conversion to SPMS: subtracting the measurement variance attributable to age would then significantly weaken the models. Therefore, ignoring effects of covariates is expedient, but not scientifically correct.
- The novelty is in using Healthy Volunteer (HV) age and sex adjustment: by subtracting only effect of healthy aging and healthy sex effects, we effectively retain MS-specific effects. We refer the reviewer back to Figure 2: GDF15, a biomarker of mitochondrial dysfunction strongly increases with age both in HD and MS but does so with different slopes. If we did not have HD data and simply use age in MS patients to regress out covariate effect of age, such age adjusted GDF15 would no longer increase with MS duration. We would conclude (falsely) that mitochondrial dysfunction does not increase with MS duration. Instead, subtracting only effect of HD age, we see that GDF15 has residual positive correlation with age/MS duration in MS cohort. So, we conclude (correctly) that mitochondrial dysfunction increases with natural aging process in CNS but does so MORE in MS patients.

Line 521-5222: here is probably missed the result showing comparison of the biological markers included in the different models. In the paragraph from line 320 on, I see mention of different numbers of aptamer ratios for each model, but not whether these were overlapping between the modes. This statement should be sustained better and the results elaborated.

- There is 99 somamer ratios used by the three models, 97 of those are unique, 2 somamer ratios are shared between the two models predicting MSDSS at baseline and follow-up. We now added this information to Results section (lines 359-360).
- Even though the individual biomarkers selected by each model may be different, the pathways reflected by these biomarkers are greatly overlapping.

Line 536, the elaboration of notch in the discussion is a novel result, in order words, such details are not elaborated well in the narrative of the results and should be.

- Identification of NOTCH signalling pathway is listed as a new finding in the results section and discussed in the Discussion section.

The discussion of the clotting/coagulation mechanism is hampered by not stating whether this mechanism was positively or negatively affected, reflected in which specific proteins, and thus by some more granularity in the discussion, and similar to the notch discussion, more details in the results. And further more clues, e.g. from the literature, what this clotting mechanism may mean in CSF. Why are these clotting proteins present in CSF? What would their role be?

- Figure 5 contains the data the reviewer is asking for: Positive associations between complement and coagulation pathways and MS severity means that increased levels of biomarkers are associated with faster MS progression. We have also likewise already discussed the origin of complement proteins: “For example, increased CSF levels of early complement proteins may not reflect their blood origin, but rather a proinflammatory, toxic response of microglia and astrocytes...” We are also citing pathology studies that previously identified (and extensively discussed) increased levels of biomarkers of clotting cascade in MS brains.

I see a discrepancy in the reasoning: the results seem to inform the compartmentalisation hypothesis in MS progression. Yet, further on in this paragraph the authors seem to state that we cannot draw any conclusions, due to the cancer bias of the pathway annotation databases.

- We do not state that we cannot draw ANY conclusions; we only state that because current databases have cancer bias, some of the biological processes that have been already linked to MS severity, such as toxic astroglia¹⁵ are not correctly annotated.

Reviewer #3 (Remarks to the Author):

This is a very interesting manuscript investigating the CSF of patients with MS by proteomics using a DNA-adaptamer-based technology. The aim is to identify biomarker or signatures of biomarkers that can predict disability progression and pathophysiological differences in order to design future personalized treatments for MS patients.

The platform used could potentially identify 1305 proteins. A machine learning approach was chosen to model CNS damage and disability progression.

The major findings are that MS was not associated with accelerated parameters for aging. Furthermore, different pathways were identified in the functional enrichment analysis that correlated with disease severity and MRI atrophy, suggesting heterogeneity of the pathomechanisms between individual patients. Finally, they could discriminate between disease severity and future disability progression.

This study is innovative and important for the possible development of individualized treatments. Most conclusions seem to be supported by the presented data. However, there are a few points that require attention:

- I fully agree with the authors that one has to control and correct for many variables in such a study and a precise clinical characterization is essential. The authors have put a great emphasis to correct for age-dependent markers and gender differences. However, there are several variables that are not clear if it was corrected for or how this was dealt with:

1. Had all patients positive OCB? Or were there differences between OCB pos. and neg. patients?

- We have OCB results available for 175 out of 227 patients. Out of those, 164/175 (93.7%) patients had positive OCB. Only 11 patients were OCB negative, which is too small cohort to study separately

2. Were the data adjusted regarding the blood-CSF-barrier? Some MS patients may have a small increase in QAlb as measure for the blood-CSF-barrier. This may lead to more blood-derived proteins in the CSF and may thus skew the analysis.

- We have analyzed correlation between Albumin quotient and the exact number of contrast enhancing lesions (CEL), as well as dichotomized CEL outcome (present/absent) and didn't find any significant relationship between them:

- Similarly, we looked at relationship between albumin quotient and MS severity outcomes (MSSS, ARMSS, and MSDSS) and didn't find significant differences:

3. Differences in IgG production in the CSF have been described as predictor for progression. Were the data corrected/correlated for QIgG?

- We do not understand why the reviewer would want to correct for IgG index, when IgG index reflect biology measurable by CSF biomarkers, which does, as reviewer correctly stated, correlate with MS severity. That would be analogous to “correcting” MRI-biomarker based predictive models for brain atrophy: what would be the purpose of that? In terms of comparing effect sizes, IgG explains <7% of MS-DSS variance, so the 3 CSF-biomarker-based models clearly outperform IgG index.

- Were the HV really healthy volunteers? Why did they receive a LP?

- Yes. Recruitment of Healthy Volunteers for comprehensive clinical, imaging, and biomarker evaluation, including the spinal tap is part of our Natural

History protocol. They provide normative data for CSF measurements for which no normative data exist. The PI is one of those healthy volunteers.

- The quality control should be described in the methods. Were all the 1305 proteins reliably detected in the CSF? How were missing data/values below detection limit dealt with?

- Not all of the 1305 proteins are reliably detected in the CSF, however there is no simple method to determine which Somamers signal and which generate just noise. The implementation of our pipeline assures that Somamer ratios measuring just noise are weaned out in the iterative process and only truly signaling Somamer ratios are considered by the model.

- All patients were untreated at the time of LP and high efficacy drugs within 6 month of LP were excluded. However, the immunological effects of Alemtuzumab and probably also Rituximab/Ocrelizumab, last for longer than 6 month.

- We have 8 samples off ocrelizumab/rituximab (no alemtuzumab treated patients) – 2 in training cohort, 6 in the validation cohort. The shortest period between stopping of the drug and LP is 9 months, the longest 55.6 months). As a sensitivity analysis, the 6 patients off Rituximab/Ocrelizumab were removed from the validation cohort, with minimal effect on the model performance.

- The patients were not during relapse at the time of LP, however, it is not stated how long after the last relapse the LP was performed. This may have a great influence on the dynamics of the CSF changes. This would also have implications on the interpretation of the results. It can be assumed that after a relapse the protein composition in the CSF changes dynamically. Thus, could it be that the heterogeneity is rather a continuum, e.g. first the myeloid or complement module, then the adaptive immunity module and then the repair module?

- We have two patients that received IV steroids for relapse – 5-6 months prior to their LP.
- If the engagement of protein modules had predictable dynamics as the reviewer suggests, then we should see preferential grouping of RRMS versus PMS, which we do not see

- The patients had a rather long disease duration, even in the RRMS cohort of 4.8 and 4.2 years, respectively. An earlier timepoint, e.g., at first diagnosis of MS, would be more important since at this time the treatment decision is often made and knowledge of prognosis would be helpful.

- Our disease onset is calculated based on date of first symptom attributable to MS – not based on date of MS diagnosis.
- Most patients were diagnosed by us, and the LP included in this study was the first LP they ever had – that is why we have so many LPs in untreated stage

- Nothing is said about the treatment of the patients after LP. I would assume that many of the patients received a DMD. This of course has an effect on the development of future disability. Thus the analysis for future disability accumulation.

- Correct, most patients received DMT after their LP and we adjust for these therapy effects when calculating the therapy-adjusted CombiWISE slope as described in ¹³

- Could the different MS forms (RRMS/SPMS/PPMS) be discriminated by the profile? This was not clear for me.

- As shown in the Figure 8, 97 CSF biomarkers included in the 3 validated models of MS severity cannot differentiate MS subtypes
- However, a different set of CSF biomarker measured by same assay can distinguish between RR-MS and progressive MS, but can't differentiate PP-

MS from SP-MS as described in ⁸. This information (with citation) is included in the Introduction.

- The authors (correctly) criticize that NfL is only predictive on a group level. However, are their results predictive on a single patient basis? And have they made a comparison of their prediction model with NfL measurements?

We measured NFL levels in the serum and CSF and the predictive power of presented models greatly outperform NFL:

○ There is no significant correlation between cNFL and MS severity:

- With regard to clinical progression it would be interesting to have also information cognition and not only on physical disability. This has in particular relevance for the cortical atrophy observed in the MRI.

- We have previously published that BPFr (together with MRI T2LL) correlate with SDMT¹². However, in the same paper we show that too few patients reproducibly progress on SDMT over longitudinal follow-up of few years due to very high learning effect and high intra-individual SDMT variance in longitudinal measurements.

- Inclusion of spinal cord MRI data would also be of interest. This may explain for the dichotomy of e.g. low brain damage with high physical disability.

- Please see our general comments: we have already included multimodality data that are not published in other modeling studies of MS outcomes: surely, the reviewer would not ask MS imaging study to measure CSF (or even serum) biomarkers.

- Have any of the patients also received a brain biopsy? Could the findings of the heterogeneous pathways be linked to the heterogeneity of the neuropathology, e.g. for complement? The discrimination of type 1 and type 2 lesions according to Lucchinetti et al. from the CSF would be extremely helpful.

- None of these patients had brain biopsy

Reviewer #4 (Remarks to the Author):

This article proposes an approach to use CSF protein biomarkers and machine learning to provide insight into multiple sclerosis pathophysiology, and claims that the results can support individual clinical prognostication. Using the CSF samples of 129 untreated MS subjects (RRMS = 31, SPMS = 37, PPMS = 61) and 24 healthy individuals, the authors extracted 1305 biomarkers, performed adjustments to minimize the effects of physiological aging and sex, then trained three random forest machine learning models to find evidence of molecular pathways associated with MS severity and brain atrophy. Validation was performed on an independent data set from 64 MS subjects, with correlations to MS Disease Severity Score (at baseline and follow-up of mean 4.2 years) and “MRI Severity” (brain atrophy) as outcomes. These correlations characterized the predictive value of the biomarkers and were moderate, ranging from 0.4 to 0.6. One key biological insight was that the molecular pathways in determining MS severity may be different from the determinants of brain atrophy severity.

This paper has a number of strengths. Despite decades of research, MS remains a very difficult disease to predict, and novel techniques to further elucidate the molecular mechanisms of MS pathology and produce more accurate individualized prediction models are welcomed by the MS community. The study appears to be carefully conducted and the manuscript is clearly written. The selection of methods seems to be sound, including the use of the MS-DSS (recognizing the limitations of the EDSS for short-term prognostication), the overall proteomics framework, and the random forests with iterative variable selection. The biological insights on the molecular pathways governing MS severity and brain atrophy appear to be significant and novel (although I am not an expert in this area).

However, there are also a number of significant weaknesses. The number of subjects used for this study is simply too small to make any claims on the results providing “reliable prognostic information”. **At best this is a proof-of-concept study.** The authors partially acknowledge this by stating “The sample sizes of the training and validation cohorts may seem small if judged by conventional, EDSS-based outcomes.”, but MS is such a heterogeneous disease that it is imperative that the study cohort is biologically and clinically representative of the general MS population, and there is no statistical evidence that this is true.

- Please see our general comments and added results and Discussion sections. The p-values in the independent validation cohort were some of the lowest p-values reported in the meta-analysis of 302 MS modeling papers, while the effect sizes, especially independent cohort validated effect sizes are the strongest reported.
- We also added new patients to have total of 98 subjects in the independent validation cohort and this did not change the conclusions of the paper.

Another problem is that some key methodological details are missing, particularly for the machine learning components. For example, no description of the unsupervised clustering

algorithm mentioned on p. 22 was provided, and the procedure for determining the hyperparameters of the random forest models was also missing.

- Description of the methods used for the unsupervised clustering algorithm has been added on in the "Statistics" section of the Material and Methods.
- Details for ntree and mtry for random forest models have been added, all previously described random forest-related methods are referenced in the manuscript, the code used to generate the random forest models has been deposited in the Github and referenced in the manuscript.

Another weakness is that the manuscript appears to downplay the importance of MRI for MS prognostication. The authors used normalized whole brain volume to represent "MRI severity", but whole brain volume represents a small fraction of the information that can be derived from imaging, especially with quantitative MRI such as magnetization transfer or myelin imaging, with the former being widely available.

- To address this point and to interpret our data in relationship of all MS modeling studies, we performed and published meta-analysis of 302 MS modeling papers. Current study outperforms all of them. If the reviewer can cite a specific MRI paper that predicts MS severity or future rates of MS disability progression in the independent validation cohort than we would gladly compare our results to such paper.

Also, it is well known that cross-sectional brain volume measurements, such as the brain parenchymal fraction employed in this study, are not optimal for longitudinal studies, and registration-based methods such as FSL SIENA are much more precise and accurate for determining change.

- We are not using longitudinal brain atrophy data – we use age-regressed brain parenchymal fraction as a measure of brain atrophy in a cross-sectional cohort of untreated MS patients. We have searched for and did not identify any FSL SIENA paper that outperforms current study in predicting MS severity outcomes or future rates of disability progression.

The authors suggest that CSF testing to estimate age-adjusted brain volume can obviate the need for actual MRI measurement, which is not well-founded given that image-based measurement has already established its scientific validity, practicality, and clinical utility.

- We simply cannot find this suggestion in the text of our paper. We have used this outcome NOT to obviate brain MRIs in MS patients but to gain insight into mechanisms that may cause CNS tissue destruction measured as brain atrophy.

Recommendations:

This manuscript has merit for publication, with significant modifications. The focus should be on the biological insights discovered, and claims of supporting precision medicine should be removed until a much larger study can be performed. “MRI severity” should be renamed “brain atrophy severity” and the narrow view of imaging in this study should be acknowledged. More details on the machine learning methods should be provided, so that the experiment can be potentially reproduced.

- “MRI severity” has been replaced throughout the manuscript with “Brain atrophy severity”
- More details on random forest modeling were provided in the methods; the R script used to generate the models is available for the purpose of reproducing of the study.

Minor issues:

-Use “sex” instead of “gender”; this is the current scientific standard.

- “gender” has been replaced with “sex” throughout the manuscript

-Page 7, “NeurEx-based outcomes are more accurate”; the statement is about variability, so perhaps “accurate” should be replaced by “precise”.

- “accurate” has been replaced with “precise”

-Page 17, “training cohort performance grossly overestimated performance in the new set of subjects”; In machine learning, training performance simply estimates fit of the model to the training data, and does not estimate performance in new data.

- The sentence refers to the generalizability of the effect sizes measured in the training cohort. The sentence was removed from the manuscript.

-Page 20, “correlations of MRI severity... outcomes is only weak” should be “correlations of MRI severity... outcomes are only weak”.

- The sentence has been corrected

-Page 26, There is a strange link:

(<http://www.keystonesymposia.org/index.cfm?e=web.Meeting.Program&meetingid=1654>)

- The link has been removed

Citations supporting our responses to the reviewers:

1. Freedman, L.P., Cockburn, I.M. & Simcoe, T.S. The Economics of Reproducibility in Preclinical Research. *PLoS biology* **13**, e1002165 (2015).
2. Ioannidis, J.P. Why most published research findings are false. *PLoS Med* **2**, e124 (2005).
3. Hackam, D.G. & Redelmeier, D.A. Translation of research evidence from animals to humans. *JAMA* **296**, 1731-1732 (2006).
4. Int'Hout, J., Ioannidis, J.P., Borm, G.F. & Goeman, J.J. Small studies are more heterogeneous than large ones: a meta-meta-analysis. *Journal of clinical epidemiology* (2015).
5. Ioannidis, J.P. Why most discovered true associations are inflated. *Epidemiology* **19**, 640-648 (2008).
6. Xu, Y. & Goodacre, R. On Splitting Training and Validation Set: A Comparative Study of Cross-Validation, Bootstrap and Systematic Sampling for Estimating the Generalization Performance of Supervised Learning. *J Anal Test* **2**, 249-262 (2018).
7. Liu, J., Kelly, E. & Bielekova, B. Current Status and Future Opportunities in Modeling Clinical Characteristics of Multiple Sclerosis. *Front Neurol* **13**, 884089 (2022).
8. Barbour, C. *et al.* Molecular-based diagnosis of multiple sclerosis and its progressive stage. *Ann Neurol* **82**, 795-812 (2017).
9. Kosa, P. *et al.* Development of a Sensitive Outcome for Economical Drug Screening for Progressive Multiple Sclerosis Treatment. *Front Neurol* **7**, 131 (2016).
10. Lorscheider, J. *et al.* Defining secondary progressive multiple sclerosis. *Brain* **139**, 2395-2405 (2016).
11. Kappos, L. *et al.* Predictive value of gadolinium-enhanced magnetic resonance imaging for relapse rate and changes in disability or impairment in multiple sclerosis: a meta-analysis. Gadolinium MRI Meta-analysis Group. *Lancet* **353**, 964-969 (1999).
12. Pham, L. *et al.* Smartphone-based symbol-digit modalities test reliably captures brain damage in multiple sclerosis. *NPJ Digit Med* **4**, 36 (2021).
13. Weideman, A.M. *et al.* New Multiple Sclerosis Disease Severity Scale Predicts Future Accumulation of Disability. *Front Neurol* **8**, 598 (2017).
14. Kosa, P. *et al.* Enhancing the clinical value of serum neurofilament light chain measurement. *JCI Insight* **7** (2022).
15. Masvekar, R., Phillips, J., Komori, M., Wu, T. & Bielekova, B. Cerebrospinal Fluid Biomarkers of Myeloid and Glial Cell Activation Are Correlated With Multiple Sclerosis Lesional Inflammatory Activity. *Frontiers in neuroscience* **15**, 649876 (2021).

REVIEWER COMMENTS

Reviewer #1 (Remarks to the Author):

The authors performed an extensive review of the paper, which lead to a substantial improvement of the manuscript.

Nevertheless, I still disagree with is said about brain volumetric results. Atrophy is a process of volume loss over time, which is normally quantified longitudinally.

As the authors report, it is also possible to estimate a potential volumetric loss by comparing the brain volume (parenchymal fraction) at a certain time point in a given patient to a distribution of volumes obtained in healthy subjects: but this requires to establish a normative range for age in the healthy group, which is not what the authors did in this work.

Therefore, I suggest to use the word BPF instead of brain atrophy across the entire manuscript and would recommend to accept the paper after this last modification.

Reviewer #2 (Remarks to the Author):

The authors have made a serious effort to restructure the paper and make it clearer, which was my main objection, which made it difficult to appreciate and led to many questions for the first version. I furthermore appreciate their argumentations, although sometimes slightly angry in tone, but addressing the issues with rational arguments.

I do not agree with the authors that Somascan is better than Olink though. But this is not the place to discuss this further.

Additional Reviewer #2 comments in relation to previous Reviewer #4 concerns:

The second question, whether data were adjusted for BBB function, was responded to differently than expect: they show a lack of relation of the BBB markers with the clinical progression measures. I would expect to see results showing effects of BBB damage (Q Alb or CEL in the previous question) on the Somascan results. I am very interested to see those.

About the predictive value of NfL. I was also struck by the quite bold statement that nFtL has no value on the single patient level. Even though the model of the novel SOMAScan blood biomarkers outperforms the predictive value of cNfL in this study, the authors did not show use that it works on the individual level yet. Thus, I suggest to weaken the statement of this failure of NfL as it indeed brings the suggestion that the novel model will solve this. An additional way of solving this is adding a few words to the discussion for future studies.

The question of cognition is also not answered directly. The authors mention only one test, but there are other cognitive tests as well. Stating that these tests have no longitudinal value is a bit short. In addition, there are several studies showing relevance of measuring cognition. I suggest here to spend a few words: MS is not only change in MRI, but also change in cognition.

REVIEWER COMMENTS

Reviewer #1 (Remarks to the Author):

The authors performed an extensive review of the paper, which lead to a substantial improvement of the manuscript.

Nevertheless, I still disagree with is said about brain volumetric results. Atrophy is a process of volume loss over time, which is normally quantified longitudinally.

As the authors report, it is also possible to estimate a potential volumetric loss by comparing the brain volume (parenchymal fraction) at a certain time point in a given patient to a distribution of volumes obtained in healthy subjects: but this requires to establish a normative range for age in the healthy group, which is not what the authors did in this work. Therefore, i suggest to use the word BPF instead of brain atrophy across the entire manuscript and would recommend to accept the paper after this last modification.

- We agree with the reviewer that atrophy is a process of CNS tissue loss over time and it is best to quantify it longitudinally. However, longitudinal measurements of brain volume in MS are affected by treatments, which are now applied to most patients after MS diagnosis. We have replaced “atrophy” with BPF wherever it made sense (i.e., replaced the words “brain atrophy” with “brain parenchymal fraction (BPFr) on line 85, with words “CNS tissue loss” on lines 155-156, and more on lines 158 and 160)
- However, we could not change it everywhere because the requests from different reviewers contradict each other: we already changed our original designation of the “MRI severity” outcome to “Brain atrophy severity” on the request of Reviewer #4. Renaming it to “Brain parenchymal fraction severity” would compromise the meaning of the outcome. We agree with the reviewer that it is a question of semantics and by defining exactly how this outcome is calculated the reader understands what it means.

Reviewer #2 (Remarks to the Author):

The authors have made a serious effort to restructure the paper and make it clearer, which was my main objection, which made it difficult to appreciate and led to many questions for the first version. I furthermore appreciate their argumentations, although sometimes slightly angry in tone, but addressing the issues with rational arguments.

I do not agree with the authors that Somascan is better than Olink though. But this is not the place to discuss this further.

Additional Reviewer #2 comments in relation to previous Reviewer #4 concerns:

The second question, whether data were adjusted for BBB function, was responded to differently than expect: they show a lack of relation of the BBB markers with the clinical progression measures. I would expect to see results showing effects of BBB damage (Q Alb or CEL in the previous question) on the Somascan results. I am very interested to see those.

- Previous studies analyzed individual MS-relevant biomarkers, such as chemokines or proportions of specific immune cells in paired CSF and blood samples (1, 2). These studies showed no correlations between CSF and blood levels of these biomarkers and found only CSF levels correlating with MS outcomes. There is therefore consensus in the field of neuroimmunology that adjusting levels of CSF proteins for blood levels is not useful.
- We do not want to complicate things further, because this discussion is not relevant to the current paper (i.e., we already showed that models from unadjusted CSF biomarkers predict MS severity outcomes with the largest effect sizes reported in the literature) but we understand that the reviewer may bring IgG index as a counter-argument. Unfortunately, the issue of protein transfer between blood and intrathecal compartment is much more complicated than any one of us appreciated. IgG index has been developed based on the observations in large numbers of CSF samples that included patients with and without clinically-defined CSF inflammation, but to our best knowledge, the concept on which IgG index is based has never been experimentally validated. There are already data in published literature that indicate that the concept of IgG index is not generally valid, as it assumes only transfer of IgG from blood to CSF. The administration of therapeutic monoclonal antibodies directly to CSF prove this concept incorrect: e.g., we reported that when rituximab is injected to lumbar cistern of MS patients, most of it ends up in the blood within next 12h, effectively depleting B cells in the blood (3). This means that IgG index severely underestimate the intrathecal production of IgG. In fact, we believe that transfer of most proteins between blood and intrathecal compartment is bidirectional (e.g., see NFL and GFAP) and protein-specific: i.e., for some proteins the transfer is passive, while for others, there is receptor-mediated active transfer, which may dominate one way: e.g. proteins necessary for CNS biology but not produced intrathecally are preferentially transported into CNS, while IgG is actually preferentially transported out of CNS based on the rituximab kinetics. We just want to conclude that adjusting levels of CSF proteins for blood levels may make sense for a specific protein, but because of the complexity of such bi-directional transfer as we have alluded to, it does not make sense to do it globally for all proteins. We must gain much more knowledge about protein transfer between the two compartments before the reviewer's idea can be effectively implemented.

- Finally, we don't really know how to incorporate correlations between SOMAScan and CELs into the manuscript, as this is not the manuscript topic. However, we have included Excel file showing correlation coefficients and FDR-adjusted p-values between CEL and Somascan data to satisfy reviewer's interest. As it can be seen, our SOMAScan data validate already published data in the literature.

About the predictive value of NfL. I was also struck by the quite bold statement that nFL has no value on the single patient level. Even though the model of the novel SOMAScan blood biomarkers outperforms the predictive value of cNfL in this study, the authors did not show use that it works on the individual level yet. Thus, I suggest to weaken the statement of this failure of NfL as it indeed brings the suggestion that the novel model will solve this. An additional way of solving this is adding a few words to the discussion for future studies.

- We thank the reviewer for the comment; however, we couldn't identify the above mentioned statement that "NFL has no value on a single patient level" in the manuscript or in our response to the reviewers. The data we added to the results section (lines 446-467) fully support the statement: "Thus, we conclude that CSF-biomarker-based models outperform NFL in predicting future rates of MS disability accumulation."
- In the response to reviewer's #2 comment we stated that "while sNFL correlates with MS severity on a group level, its accuracy is likely too low to be clinically meaningful on a patient level." This statement is based on published studies of clinical trials and considers reported p-values and samples size, inferring low effect size value. We truly hope that future reports of NFL measurements in MS clinical trials will include effect sizes, as it is incorrect to assume clinical utility of the test based on p-value only.

The question of cognition is also not answered directly. The authors mention only one test, but there are other cognitive tests as well. Stating that these tests have no longitudinal value is a bit short. In addition, there are several studies showing relevance of measuring cognition. I suggest here to spend a few words: MS is not only change in MRI, but also change in cognition.

- We do not show any data related to cognitive outcomes in the manuscript.
- In the response to Reviewer's #2 comment we stated that SDMT outcome is not suitable for our analysis because of its high degree of "training effect" and high measurement noise. These statements are supported by published data (4)
- We have collected PASAT data in addition to SDMT, and we have shown the very same problem with high noise of this measurement (5)
- While we agree with the reviewer that MS is not only change in MRI, but also change in cognition, uncovering biology related to cognitive decline will require better outcome measures.

References:

1. Hannikainen PA, Kosa P, Barbour C, and Bielekova B. Extensive Healthy Donor Age/Gender Adjustments and Propensity Score Matching Reveal Physiology of Multiple Sclerosis Through Immunophenotyping. *Front Neurol.* 2020;11:565957.

2. Lepennetier G, Hracsko Z, Unger M, Van Griensven M, Grummel V, Krumbholz M, et al. Cytokine and immune cell profiling in the cerebrospinal fluid of patients with neuro-inflammatory diseases. *J Neuroinflammation*. 2019;16(1):219.
3. Komori M, Lin YC, Cortese I, Blake A, Ohayon J, Cherup J, et al. Insufficient disease inhibition by intrathecal rituximab in progressive multiple sclerosis. *Ann Clin Transl Neurol*. 2016;3(3):166-79.
4. Pham L, Harris T, Varosanec M, Morgan V, Kosa P, and Bielekova B. Smartphone-based symbol-digit modalities test reliably captures brain damage in multiple sclerosis. *NPJ Digit Med*. 2021;4(1):36.
5. Kosa P, Ghazali D, Tanigawa M, Barbour C, Cortese I, Kelley W, et al. Development of a Sensitive Outcome for Economical Drug Screening for Progressive Multiple Sclerosis Treatment. *Front Neurol*. 2016;7:131.

REVIEWERS' COMMENTS

Reviewer #1 (Remarks to the Author):

I would encourage the authors to eliminate the word atrophy from the entire manuscript and substitute it with volume, as the first is not correct based on current knowledge.

As such, brain atrophy severity does not make any sense if a healthy control population or a longitudinal assessment are not available.

What you report are volumes in the form of brain parenchymal fractions not atrophy (= brain volume changes vs healthy controls or vs longitudinal follow-ups).

This is not a semantic preference but a conceptual point that is important and should be considered, in order not to decrease the value of the overall work.

After this change, I recommend the work for publication.

Cristina Granziera

Reviewer #2 (Remarks to the Author):

Great work!

Please feel free to accept it.

REVIEWERS' COMMENTS

Reviewer #1 (Remarks to the Author):

I would encourage the authors to eliminate the word atrophy from the entire manuscript and substitute it with volume, as the first is not correct based on current knowledge.

As such, brain atrophy severity does not make any sense if a healthy control population or a longitudinal assessment are not available.

What you report are volumes in the form of brain parenchymal fractions not atrophy (= brain volume changes vs healthy controls or vs longitudinal follow-ups).

This is not a semantic preference but a conceptual point that is important and should be considered, in order not to decrease the value of the overall work.

After this change, I recommend the work for publication.

Cristina Granziera

Response: We have changed “Brain atrophy” phrase with “Brain volume deficit” abbreviated as “BVD” throughout the main text, figures, tables, and supplementary information.

Reviewer #2 (Remarks to the Author):

Great work!

Please feel free to accept it.

Response: Thank you.